# Stochastic Sparse Attention for Memory-Bound Inference

**Kyle Lee** [1]  **Corentin Delacour** [1]  **Kevin Callahan-Coray** [1]  **Kyle Jiang** [1]  **Can Yaras** [2]  **Samet Oymak** [2]
**Tathagata Srimani** [3]  **Kerem Y. Camsari** [1]

## Abstract

Autoregressive decoding becomes bandwidth-limited at long contexts, as generating each token requires reading all $n_k$ key and value vectors from KV cache. We present Stochastic Additive No-mulT Attention (SANTA), a method that sparsifies value-cache access by sampling $S \ll n_k$ indices from the post-softmax distribution and aggregates only those value rows. This yields an unbiased estimator of the post-softmax value aggregation while replacing value-stage multiply-accumulates with gather-and-add. We introduce stratified and systematic sampling to design variance-reduced, GPU-friendly variants. Evaluated on Llama-3.1-8B-Instruct at 32k-token contexts, $S^2$ANTA matches baseline accuracy while achieving up to $1.5\times$ decode-step attention-kernel speedup over FlashInfer and FlashDecoding on an NVIDIA RTX 6000 Ada. In batched long-context generation, these kernel gains translate to up to $1.25\times$ end-to-end decode-latency speedup. Finally, we propose Bernoulli $qK^\mathsf{T}$ sampling as a complementary technique to sparsify the score stage, reducing key-feature access through stochastic ternary queries. Both methods are complementary to upstream quantization, low-rank projection, KV-cache compression, and KV-cache selection methods. Together, they point toward sparse, multiplier-free, and energy-efficient inference. We open-source our kernels at: https://github.com/OPUSLab/SANTA.git

---

[1]Department of Electrical and Computer Engineering, University of California, Santa Barbara, CA, USA [2]Department of Electrical and Computer Engineering, University of Michigan, Ann Arbor, MI, USA [3]Department of Electrical and Computer Engineering, Carnegie Mellon University, Pittsburgh, PA, USA. Correspondence to: Kyle Lee <kylelee@ucsb.edu>.

*Proceedings of the 43$^{rd}$ International Conference on Machine Learning*, Seoul, South Korea. PMLR 306, 2026. Copyright 2026 by the author(s).

## 1. Introduction

Transformers (Vaswani et al., 2017) underpin modern language models (OpenAI, 2023; Touvron et al., 2023; Gemini Team et al., 2023). As deployments increasingly rely on long contexts, autoregressive decoding enters a regime where throughput is often limited by memory bandwidth: generating each new token repeatedly streams the Key–Value (KV) cache for all prior tokens. For example, for Llama-3.1-8B-Instruct at a 32k-token context, streaming bf16 keys and values is on the order of $\sim$128 MB *per layer per generated token*, and KV memory demands scale linearly with context length and batch size.

A broad literature mitigates this bottleneck from several angles. KV-cache quantization/compression reduces bytes per element (Liu et al., 2024; Hooper et al., 2024); cache-management policies reduce the number of retained tokens (Tang et al., 2024; Zhang et al., 2023); and structured sparsity or candidate selection reduces attention cost (Child et al., 2019; Zaheer et al., 2020; Beltagy et al., 2020; Wang et al., 2020; Kitaev et al., 2020; Chen et al., 2025). Architectural changes such as grouped-query attention (GQA) reduce KV footprint by sharing keys/values across heads (Ainslie et al., 2023), and decoding kernels improve IO locality for *exact* attention (Dao et al., 2023; Ye et al., 2025). However, even with optimized exact kernels, long-context decoding still incurs a persistent cost: each step touches essentially the full KV state. In contrast to methods that decide which tokens or cache blocks are retained, evicted, or reconstructed, our goal is to reduce the cost of the value aggregation operator over whatever KV state is made available at decode time.

We study a complementary inference-time lever: *reducing the amount of KV data accessed per decode step* without permanently discarding cache contents. Our primary focus is the post-softmax value aggregation, where dense attention computes $AV$ by multiplying attention weights by all $n_k$ value rows. We introduce **S**tochastic **A**dditive **N**o-mul**T** **A**ttention (**SANTA**), which samples $S \ll n_k$ indices from the post-softmax distribution and aggregates only those sampled rows of $V$ via gather-and-add. This yields an unbiased estimator of the post-softmax value aggregation while reducing value-cache row reads per query from $n_k$ to $S$ (and eliminating value-stage multiplications in favor of additive

accumulation). We further propose variance-reduced variants ($\mathbf{S^2ANTA}$) based on stratified and systematic sampling that bolster accuracy and admit GPU-friendly implementations. Since sparsifying $V$ reads yields the clearest bandwidth benefit in decoding, our systems evaluation targets decode-step acceleration at long contexts.

On modern GPUs, these changes translate into measurable kernel-level gains: our $S^2$ANTA CUDA kernels achieve up to $\mathbf{1.5\times}$ **decode-step attention-kernel speedup** over FlashInfer and FlashDecoding at 32k-token contexts while matching baseline accuracy on long-context prompts (Section 3).

Finally, we propose **Bernoulli** $qK^\mathsf{T}$ **sampling** as a complementary technique for the score stage: it represents query elements as Bernoulli variables to form sparse ternary queries in $\{-1, 0, +1\}$, yielding an unbiased estimator of $qK^\mathsf{T}$ and inducing feature-sparse key access during decoding. In this work, our implemented kernels and measured speedups are driven primarily by value-stage sparsification; Bernoulli $qK^\mathsf{T}$ is presented as a complementary mechanism that can be combined with SANTA to sparsify both branches.

**Contributions.** Concretely, we: (i) introduce SANTA and variance-reduced $S^2$ANTA as unbiased estimators of the post-softmax value aggregation that reduce value-cache reads from $n_k$ to $S$; (ii) design GPU kernels that parallelize sampling and sparse accumulation, demonstrating up to $1.5\times$ decode-step attention-kernel speedup and up to $1.25\times$ end-to-end decode-latency speedup in batched long-context generation while matching baseline long-context accuracy; (iii) empirically characterize tunable accuracy–efficiency trade-offs on GSM8K, MMLU, and long-context benchmarks; and (iv) propose Bernoulli $qK^\mathsf{T}$ sampling as a complementary score-stage estimator that can reduce key-feature access during decoding.

## 2. Stochastic Additive No-mulT Attention (SANTA)

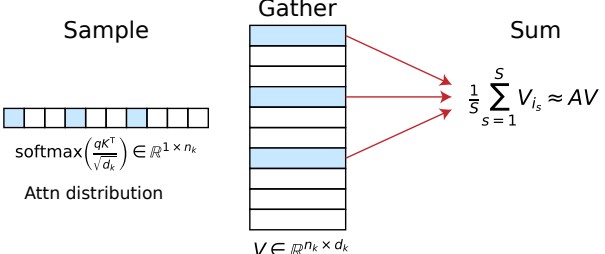

*Figure 1.* SANTA requires $S \ll n_k$ memory accesses to sampled rows of $V$, eliminating multiplications following the softmax operation. Normalization by $S$ is a bit shift if $S$ is a power of 2.

In principle, SANTA can be employed for the prefill or decode stages. Since the benefit of sparse memory access becomes apparent in decode, we present SANTA as a decoding algorithm. In autoregressive decoding, we consider a single query vector $q \in \mathbb{R}^{1 \times d_k}$. Scaled dot-product attention (SDPA) is expressed as a function of the key matrix and value matrix:

$$\mathrm{Attn}(q, K, V) = \mathrm{softmax}\left(\frac{qK^\mathsf{T}}{\sqrt{d_k}}\right) V = AV, \quad (1)$$

where $K \in \mathbb{R}^{n_k \times d_k}$ is the key matrix, $V \in \mathbb{R}^{n_k \times d_k}$ is the value matrix, and $A \in \mathbb{R}^{1 \times n_k}$ represents the attention scores as a probability distribution.

To approximate $AV$, we treat $A$ as a categorical distribution over keys and sample $S$ indices i.i.d. from it (with replacement). For query vector $q$, we fetch the corresponding $S$ rows of $V$ and average them. This yields an estimate of the attention output using only row indexing and addition.

As an illustrative example, consider number of keys $n_k = 3$. We sample $S = 3$ one-hot attention vectors from the categorical distribution $A$:

$$\widehat{AV} = \frac{1}{3}\left(\begin{bmatrix} 1 & 0 & 0 \end{bmatrix} + \begin{bmatrix} 1 & 0 & 0 \end{bmatrix} + \begin{bmatrix} 0 & 0 & 1 \end{bmatrix}\right) V, \ (2)$$

which simplifies by distributivity of matrix multiplication over addition:

$$\begin{aligned} \widehat{AV} &= \frac{1}{3}\left(\begin{bmatrix} 1 & 0 & 0 \end{bmatrix} V + \begin{bmatrix} 1 & 0 & 0 \end{bmatrix} V + \begin{bmatrix} 0 & 0 & 1 \end{bmatrix} V\right) \\ &= \frac{1}{3}\left(V_1 + V_1 + V_3\right). \end{aligned}$$
$$(3)$$

where each $V_i \in \mathbb{R}^{1 \times d_k}$ is a sampled row of $V$. The one-hot attention vectors in Eqs. 2, 3 are only for illustrative purposes; they represent an indexing and gather operation, and are not materialized as vectors in memory.

Finally, SANTA can be summarized as:

$$\widehat{AV} = \frac{1}{S}\sum_{s=1}^{S} V_{i_s} \approx AV, \quad (4)$$

where $i_s$ is a sampled index and each $V_{i_s}$ denotes a gathered row from $V$. Multiplications in the value stage have therefore been eliminated in favor of sampling, indexing and addition. If $S$ is chosen as a power of 2 (and the model has a fixed point representation), normalization may be implemented as a bit-shift. Fig. 1 illustrates SANTA's memory access benefits for autoregressive decoding: SANTA sparsely reads only $S \ll n_k$ sampled rows of $V$ and computes an additive average without multiplications after the softmax.

We provide a formal mathematical description of SANTA in Appendix A. SANTA is an unbiased estimator of the attention function whose variance scales as $1/S$ (Appendix A.1,

A.2). A dimension-free high-probability concentration bound for SANTA is deferred to Appendix B.

## 2.1. $S^2$ANTA: Stratified and Systematic Sampling

$S^2$ANTA reduces variance via stratified sampling: dividing the CDF into $S$ equal probability mass intervals and sampling once per interval for better coverage of the attention distribution. We devise two variants: independent stratified sampling ($\boldsymbol{S^2}$ANTA-strat) and systematic sampling ($\boldsymbol{S^2}$ANTA-sys).

The mechanics of $S^2$ANTA-strat remain identical to default SANTA, except the $S$ samples are drawn per stratum instead of iid from the full distribution. Systematic sampling ($\boldsymbol{S^2}$ANTA-sys) similarly splits the attention distribution CDF into $S$ equal probability mass partitions. However, whereas independent stratified sampling draws $S$ random offsets for each of $S$ strata, systematic sampling draws a single offset $U$ and applies the same offset to all strata, effectively drawing $S$ samples with a single random number.

**Construction.** Fix a query $q$. Let $A_q = (p_{q1}, \ldots, p_{qn_k})$ be the attention weights and $F_q$ the associated CDF on $[0, 1)$. Partition $[0, 1)$ into $S$ equal intervals

$$I_m := [m/S, (m+1)/S), \qquad m = 0, \ldots, S - 1.$$

Define two schemes:

- **Independent stratified ($S^2$ANTA-strat).** Draw $T_m \sim \text{Unif}(I_m)$ *independently* for $m = 0, \ldots, S - 1$, set $J_m := F_q^{-1}(T_m)$, and output $\widehat{AV}_q^{\text{ind}} := \frac{1}{S} \sum_{m=0}^{S-1} V_{J_m}$.

- **Systematic ($S^2$ANTA-sys).** Draw a single $U \sim \text{Unif}([0, 1/S))$ and take thresholds $T_m := U + m/S$. With $J_m := F_q^{-1}(T_m)$, output $\widehat{AV}_q^{\text{sys}} := \frac{1}{S} \sum_{m=0}^{S-1} V_{J_m}$.

$\boldsymbol{S^2}$ANTA-strat and $\boldsymbol{S^2}$ANTA-sys are unbiased estimators of attention (see Appendix A.4). Only $\boldsymbol{S^2}$ANTA-strat has a theoretical guarantee of variance reduction compared to default SANTA (see Appendix A.5). Despite lacking a variance reduction guarantee, we validate that $\boldsymbol{S^2}$ANTA-sys exhibits comparable performance to $\boldsymbol{S^2}$ANTA-strat on reasoning benchmarks and long-context benchmarks. $\boldsymbol{S^2}$ANTA-sys demonstrates similar variance reduction to $\boldsymbol{S^2}$ANTA-strat in practice (see Appendix D). Systematic sampling with $\boldsymbol{S^2}$ANTA-sys is appealing because it draws samples using one random number instead of $S$ random numbers, which may be easier to implement in hardware.

## 2.2. Theoretical Operations and Memory Accesses

By virtue of SANTA's sparsity, in autoregressive decoding, SANTA demonstrates a $S/n_k$ reduction in post-softmax additions and reads to rows of $V$, where $S$ is the sample budget and $n_k$ is the number of keys. SANTA removes post-softmax multiplications in favor of additive accumulation (see Appendix E for analysis of decode-step operations and memory accesses).

In prefill, while similar arguments hold for SANTA's reduced additions and removed multiplies, there is no longer sparse memory access to the value matrix $V$. Each of $n_q = n_k$ queries may each sample $S$ rows of $V$, therefore requiring $\gg S$ rows of $V$ when the union across all queries is considered (see Appendices F, G for prefill analysis).

# 3. GPU Kernels for Variance-Reduced SANTA

Section 2 introduced SANTA and variance-reduced sampling rules (stratified/systematic), which specify *what* to sample to form an unbiased estimator of the value-stage aggregation $AV$. In this section we focus on *how* to realize these samplers efficiently at decode time on GPUs: we introduce two kernel strategies, $S^2$ANTA-prop and $S^2$ANTA-flash, which differ in how they allocate samples across tiles (global proportional allocation vs. speculative per-tile sampling with deferred normalization). Both kernels instantiate the systematic $S^2$ANTA sampling rule introduced in Section 2.1; the suffixes "prop" and "flash" refer to GPU execution strategies rather than distinct mathematical estimators. Thus, Fig. 4 and Table 1 evaluate GPU-engineered versions of $S^2$ANTA-sys.

## 3.1. Parallelizing Stochastic Attention on GPUs

Efficient implementation on GPUs presents a unique systems challenge: parallelizing the sampling control flow. Standard attention (SDPA) is amenable to "split-KV" strategies (e.g., FlashDecoding (Dao et al., 2023)) because the reduction of values is associative: partial sums can be computed locally and merged later. Sampling, however, creates a sequential dependency: determining *which* rows of $V$ to access generally requires a global Cumulative Distribution Function (CDF). A naive implementation would require a sequential pass over the global probability distribution to draw sample indices, leaving the GPU compute units idle and preventing parallel access to the value matrix.

To unlock parallel $V$-access, we must determine the number of samples per tile *before* or *independently* of the full global reduction. We propose two strategies to break this dependency:

1. **Proportional Sample Allocation ($S^2$ANTA-prop):** We pre-calculate the probability mass of each tile to

deterministically assign sample budgets. This requires a global barrier but ensures exact load balancing.

2. **Speculative Local Sampling ($S^2$ANTA-flash):** Tiles sample independently with uniform budgets, speculating that all tiles are equally important, followed by a post-hoc re-weighting. This removes the barrier but introduces sample waste.

We implement both strategies and evaluate their decode-step kernel latency and long-context accuracy at 32k-token prompts in Section 3.4.

### 3.2. $S^2$ANTA-Prop: Exact Budget Allocation via Global Sync

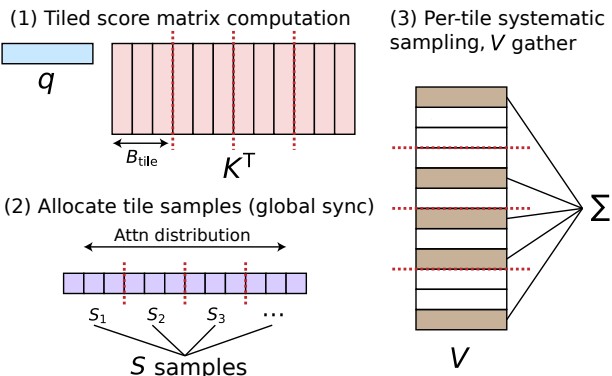

*Figure 2.* $S^2$ANTA-prop kernel overview.

$S^2$ANTA-prop (Figure 2) breaks the sequential sampling dependency by resolving the global partition function $Z$ in a lightweight initial pass.

The algorithm proceeds in three stages. The key dimension $n_k$ is partitioned into tiles of length $B_{\text{tile}}$.

1. *Pass 1 (score stash & tile stats):* We compute attention scores exactly. The exponentiated scores (or a normalized equivalent) are written to global memory. Crucially, since the scores are scalars ($1 \times n_k$) while the values are vectors ($n_k \times d_k$), this stash consumes negligible bandwidth ($1/d_k$) compared to a full value fetch. We also compute and write local partition functions $Z_{\text{tile}}$.

2. *Global budget allocation:* A separate kernel sums the tile statistics to find the global $Z$. It then assigns a specific sample count to each tile: $S_{\text{tile}} \propto S \cdot (Z_{\text{tile}}/Z)$.

3. *Pass 2 (parallel gather):* Using the stashed scores and the assigned $S_{\text{tile}}$, each tile systematically samples indices and accumulates rows from $V$. Tiles with low probability mass are assigned $S_{\text{tile}} = 0$, allowing them to skip the expensive $V$-read entirely.

Comprehensive pseudocode is in Appendix P.

### 3.3. $S^2$ANTA-Flash: Speculative Sampling with Deferred Normalization

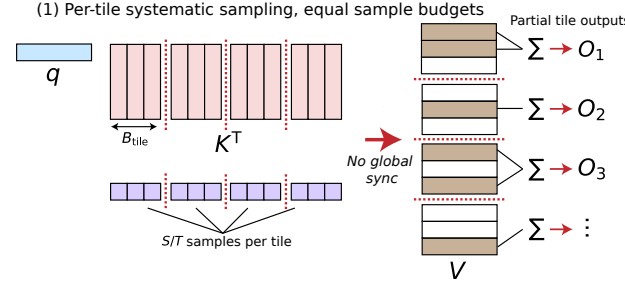

*Figure 3.* $S^2$ANTA-flash kernel overview.

To eliminate the latency of the global synchronization barrier, $S^2$ANTA-flash (Figure 3) adopts a speculative approach structurally similar to FlashDecoding.

Instead of pre-calculating the global budget, each tile is allocated a fixed, uniform budget $S_{\text{tile}} \approx S/T$, where $T$ is the number of tiles. Each tile samples locally *as if* it contained the entire probability mass. A second reduction kernel then computes the true global partition function $Z$ and down-weights the partial outputs of tiles that had low probability mass.

This method maximizes parallelism by removing the global barrier but suffers from *sample waste*: tiles with low probability mass still consume compute and bandwidth to draw samples that are effectively down-weighted during the final merge. In our 32k-token evaluation (Section 3.4), matching SDPA accuracy requires substantially larger budgets for $S^2$ANTA-flash; the accuracy-matching operating point is $S=2048$ for $S^2$ANTA-flash versus $S=128$ for $S^2$ANTA-prop (Table 1). See Appendix Q for full pseudocode.

### 3.4. Kernel Results at Long Contexts (32k)

We evaluate decode-step attention *kernel latency* together with end-task *accuracy* in the long-context decoding regime. Within this section, SDPA is used for prefill and only the *decode step* uses $S^2$ANTA. Kernel timings are reported for a single decoding step on an NVIDIA RTX 6000 Ada GPU (batch size 1) using bf16 Q/K/V tensors with Llama-3.1-8B-Instruct (Meta AI, 2024) tensor shapes; we compare against optimized FlashInfer and FlashDecoding backends. The measurement protocol and tensor layouts are provided in Appendix R.

To make latency comparisons meaningful, we select one operating point per kernel family as the *smallest* sample budget $S$ that matches SDPA accuracy on 32k-token long-context

tasks. This yields $S=128$ for $S^2$ANTA-prop and $S=2048$ for $S^2$ANTA-flash. These are exactly the configurations that deliver $\approx 1.50\times$ decode-step attention-kernel speedup in Fig. 4 while matching SDPA within confidence intervals in Table 1.

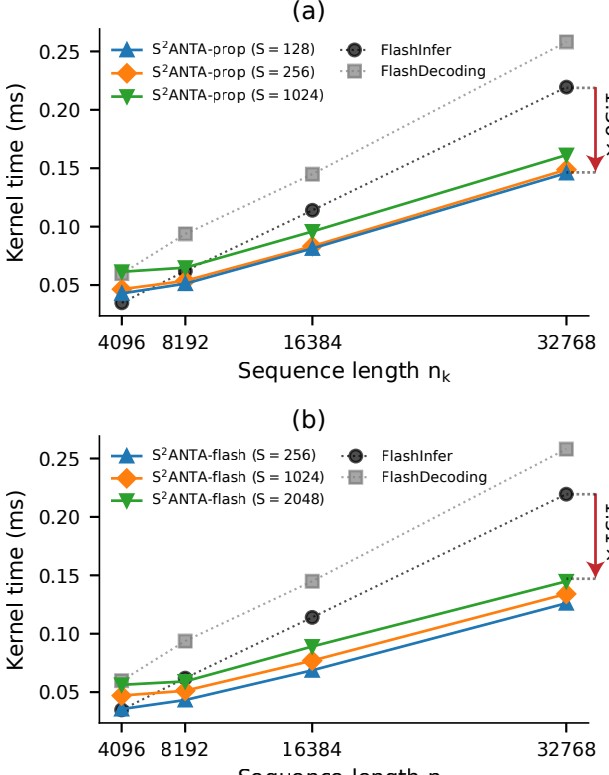

*Figure 4.* $S^2$ANTA kernel latency on Llama 8B tensor shapes for 1 decoding step. (a) $S^2$ANTA-prop demonstrates a $1.50\times$ speedup relative to FlashInfer at a 32k-token context. (b) $S^2$ANTA-flash similarly exhibits a $1.51\times$ speedup. Both operating points ($S = 128$ for prop, $S = 2048$ for flash) correspond to configurations that recover baseline accuracy (Table 1).

We next verify that these speedup points preserve long-context accuracy. We evaluate on tasks from RULER (Hsieh et al., 2024) using 32,768-token prompts: frequent-word extraction (FWE), needle-in-a-haystack (NIAH), single-hop QA (QA$_1$, derived from SQuAD (Rajpurkar et al., 2018)), and multi-hop QA (QA$_2$, derived from HotpotQA (Yang et al., 2018)). Results are shown in Table 1.

We further evaluate the same accuracy-matching operating points on LongBench v2 (Bai et al., 2025), a broader long-context reasoning benchmark. Results are shown in Table 2. Across three runs, prompts longer than 32k tokens are truncated to 32k following the repository protocol. In this 30k-token regime, $S^2$ANTA-prop with $S=128$ and $S^2$ANTA-flash with $S=2048$ match SDPA within bootstrap confidence intervals, using the same operating points where Fig. 4 reports $\approx 1.5\times$ decode-step attention-kernel speedup.

*Table 1.* Accuracy of Llama 8B on 4 long-context tasks with a 32,768-token prompt. SDPA is used in prefill. Tasks: frequent word extraction (FWE), needle-in-a-haystack (NIAH), single-hop QA (QA$_1$), multi-hop QA (QA$_2$). $S$: sample budget. Accuracy shows 95% bootstrap confidence intervals.

| Kernel | $S$ | FWE | NIAH | QA$_1$ | QA$_2$ |
|---|---|---|---|---|---|
| $S^2$**ANTA-prop** | 128 | **95.40±1.10** | **98.25±0.62** | **64.40±4.50** | **60.20±4.20** |
| $S^2$ANTA-prop | 256 | 95.47±1.10 | 98.50±0.57 | 63.40±4.10 | 60.60±4.00 |
| $S^2$ANTA-prop | 1024 | 95.40±1.07 | 98.40±0.60 | 64.20±4.00 | 59.60±4.40 |
| $S^2$ANTA-flash | 256 | 66.20±2.40 | 88.95±1.43 | 63.00±4.20 | 57.20±4.30 |
| $S^2$ANTA-flash | 1024 | 91.60±1.43 | 98.10±0.62 | 62.20±4.10 | 61.00±4.30 |
| $S^2$**ANTA-flash** | 2048 | **94.13±1.13** | **98.25±0.62** | **64.60±4.20** | **60.00±4.40** |
| **SDPA (baseline)** | — | **95.60±1.00** | **98.35±0.62** | **64.00±4.10** | **58.80±4.10** |

*Table 2.* LongBench v2 accuracy for Llama 8B at the main accuracy-matching operating points. $S$: sample budget. Accuracy shows 95% bootstrap confidence intervals.

| Kernel | $S$ | Acc. (%) | Avg. scored tok. |
|---|---|---|---|
| $S^2$**ANTA-prop** | 128 | **28.098±1.028** | **29998.4** |
| $S^2$**ANTA-flash** | 2048 | **28.827±3.422** | **29996.4** |
| **SDPA (baseline)** | — | **28.363±1.509** | **29996.0** |

At these operating points, the speedups are driven primarily by reduced memory traffic from sparsifying reads of the value matrix $V$. For $S^2$ANTA-prop, $S=128$ corresponds to extremely sparse value access at 32k contexts; even accounting for the worst-case union across Llama 8B's four-query GQA group, this is $\lesssim 1.56\%$ of value rows.

$S^2$ANTA-flash attains comparable speedup, but requires a larger total sample budget because uniform per-tile sampling expends samples on low-mass tiles before the deferred renormalization step ("sample waste").

Appendix H compares against top-$k$ at matched budgets $k = S$; top-$k$ is competitive on short-context GSM8K, but $S^2$ANTA variants outperform top-$k$ across nearly all 8k-token long-context tasks, with the largest gaps on multi-hop QA.

These timings isolate the decode-step attention kernel. We next verify that the kernel-level gain survives integration into a full generation loop, where prefill, MLPs, normalization, projections, and framework overheads remain unchanged.

The full LongBench v2 budget sweep, additional 8k-token kernel-accuracy results, and relative numerical error/tile size analyses are available in Appendices I, S, and U, respectively.

### 3.5. End-to-End Batched Decoding

We next measure end-to-end decode latency by integrating our kernels into a batched generation loop. The experiment uses Llama-3.1-8B-Instruct on an NVIDIA RTX 6000 Ada

GPU with bf16 model weights and KV cache. We use representative QA prompts truncated to 30k tokens, batch size 6, and decode 2048 new tokens per prompt. Prefill is kept dense for all methods; $S^2$ANTA is applied only during autoregressive decoding. We use the same accuracy-matching operating points as in Section 3.4: $S = 2048$ for $S^2$ANTA-flash and $S = 128$ for $S^2$ANTA-prop.

*Table 3.* End-to-end batched generation latency for Llama 8B on RTX 6000 Ada. Prompts are truncated to 30k tokens, batch size is 6, and each prompt decodes 2048 new tokens. Prefill remains dense; $S^2$ANTA is applied only during decode. Results are averaged over 9 batches.

| Method | $S$ | Prefill (s/batch) | Decode (s/batch) | ms / output token | Decode speedup |
|---|---|---|---|---|---|
| FlashAttn-2 | – | 56.69 | 101.22 | 8.237 | 1.000× |
| $S^2$ANTA-flash | 2048 | 56.61 | 80.97 | 6.589 | 1.250× |
| $S^2$ANTA-prop | 128 | 57.03 | 82.36 | 6.702 | 1.229× |

Table 3 shows that the decode-step kernel gains translate to end-to-end generation: $S^2$ANTA-flash achieves a $1.25\times$ decode-latency speedup and $S^2$ANTA-prop achieves a $1.229\times$ speedup relative to FlashAttention-2 (Dao, 2024). The prefill times remain essentially unchanged, as expected, since these measurements use dense prefill and replace only the decode-time softmax–$V$ aggregation. Thus, SANTA is best viewed as a specialized backend for long-context, decode-heavy workloads rather than a prefill accelerator.

The observed end-to-end speedup is consistent with a simple Amdahl-style bandwidth model. SANTA modifies only the decode-time softmax–$V$ aggregation, so its realized benefit is bounded by the share of decode bandwidth spent on the KV pathway. In bf16, for one batched decode step, the Llama 8B weights contribute roughly 15 GB of traffic, while the KV cache for a batch of six 30k-token prompts contributes roughly 24 GB. Combining this bandwidth share with the measured long-context attention-kernel speedup $s_{KV} \approx 1.5$ predicts $1/((15/39) + (24/39)/1.5) \approx 1.26\times$, close to the observed $1.25\times$ decode speedup in Table 3. This also explains why longer contexts and larger batches should increase SANTA's marginal benefit, while shorter contexts, prefill-heavy workloads, or aggressive KV-cache quantization may reduce it; model-weight quantization may have the opposite effect by increasing the relative share of KV traffic.

Our measured kernel and end-to-end speedups are on RTX 6000 Ada GPUs. Because SANTA reduces KV-cache traffic, we expect the qualitative benefit to remain relevant in bandwidth-bound decode on datacenter GPUs such as A100/H100, but exact speedups will depend on memory bandwidth, cache hierarchy, batching, and kernel integration.

### 3.6. Energy Considerations

Beyond latency, $S^2$ANTA's design offers two complementary paths to energy reduction. First, sparse memory access reduces value-cache traffic from $n_k$ to $S$ rows per query; at accuracy-matching operating points, this corresponds to $>90\%$ reductions in value-stage memory access at 32k contexts. Second, replacing dense multiply-accumulate with gather-and-add eliminates multiplication operations entirely. While our current GPU kernels primarily exploit the first benefit, hardware architectures optimized for sparse, adder-centric computation could realize both. These characteristics make $S^2$ANTA a natural fit for emerging AI accelerators where memory bandwidth and multiplier energy are primary constraints.

## 4. General Reasoning & Accuracy Verification

Having established the long-context decode regime in Section 3, we additionally test SANTA, $S^2$ANTA-strat, and $S^2$ANTA-sys as mathematical constructions on GSM8K, MMLU, and 8k-token long-context tasks using PyTorch reference implementations. These experiments apply SANTA in both prefill and decoding, so we use the short-context results primarily as accuracy checks rather than speedup claims; sparse value-cache memory savings are most meaningful during autoregressive decoding. HumanEval code-generation results are reported in Appendix T.

### 4.1. GSM8K

*Table 4.* GSM8K accuracy and average context length (prompt + answer) for Llama-3.1-8B-Instruct. $S$: SANTA sample budget. Accuracy shows 95% bootstrap confidence intervals.

| | $S^2$ANTA-sys | | $S^2$ANTA-strat | | SANTA | |
|---|---|---|---|---|---|---|
| $S$ | Acc. (%) | Tok. | Acc. (%) | Tok. | Acc. (%) | Tok. |
| 2 | 1.11±0.34 | 1071 | 1.01±0.32 | 1071 | 1.26±0.34 | 1075 |
| 4 | 1.67±0.40 | 569 | 1.54±0.38 | 611 | 1.57±0.37 | 825 |
| 8 | 2.10±0.45 | 340 | 1.74±0.40 | 325 | 1.44±0.38 | 207 |
| 16 | 44.63±1.58 | 349 | 39.12±1.50 | 352 | 5.51±0.72 | 350 |
| 32 | 68.59±1.42 | 339 | 67.00±1.48 | 343 | 38.26±1.43 | 348 |
| 64 | 76.42±1.34 | 341 | 74.43±1.31 | 343 | 63.63±1.49 | 346 |
| 128 | **77.33±1.34** | 342 | 75.64±1.33 | 341 | 70.23±1.42 | 341 |
| 256 | 77.56±1.34 | 343 | **78.17±1.28** | 343 | 75.61±1.42 | 342 |
| SDPA (baseline) | | | | | **78.06±1.33** | 344 |

We consider Llama-3.1-8B-Instruct ("Llama 8B") (Meta AI, 2024) on the GSM8K dataset as a mathematical reasoning benchmark (Cobbe et al., 2021; OpenAI, 2023). Table 4 provides both accuracy and the average number of tokens (prompt + answer). Because the compute and memory access costs of SANTA are contextualized relative to the sequence length $n_k$, these sequence lengths justify the regimes where $S \ll n_k$ and SANTA is, in principle, cheaper than SDPA in attention's value stage.

Default SANTA exhibits respectable performance, but variance-reduced $S^2$ANTA variants demonstrate superior performance across every sample budget $S$. $S^2$ANTA implementations approach the accuracy of full SDPA (within 1%) while $S$ remains shorter than the sequence length $n_k$, translating to memory-access savings and FLOP energy savings. In deployment settings where LLM users do not require maximum model performance, the sample budget $S$ can be tuned to save computation cost. For instance, $S^2$ANTA-sys achieves 76.42% accuracy at $S = 64$, which is about 19% of the average sequence length of $n_k = 341$ tokens. For a modest accuracy trade-off compared to SDPA's 78.06%, $S^2$ANTA-sys in principle costs 19% of the additions, 19% of the $V$-matrix accesses, and *no multiplies*.

We provide additional DeepSeek-R1-Distill-Qwen-7B (DeepSeek-AI, 2025) (DeepSeek 7B) results and comparisons to sparse top-k attention in Appendix H.

### 4.2. MMLU

*Table 5.* MMLU accuracy and average context length (prompt + answer) for Llama-3.1-8B-Instruct. $S$: SANTA sample budget. Accuracy shows 95% bootstrap confidence intervals.

| | $S^2$ANTA-sys | | $S^2$ANTA-strat | | SANTA | |
|---|---|---|---|---|---|---|
| $S$ | Acc. (%) | Tok. | Acc. (%) | Tok. | Acc. (%) | Tok. |
| 2 | 24.52±1.22 | 1141 | 24.28±1.21 | 1141 | 24.89±1.27 | 1144 |
| 4 | 24.49±1.18 | 774 | 24.65±1.26 | 807 | 25.13±1.20 | 961 |
| 8 | 24.47±1.23 | 292 | 25.63±1.23 | 286 | 25.06±1.23 | 284 |
| 16 | 34.14±1.39 | 353 | 32.46±1.32 | 350 | 26.24±1.26 | 313 |
| 32 | 45.48±1.45 | 403 | 43.78±1.50 | 398 | 33.81±1.34 | 354 |
| 64 | **48.70±1.39** | 418 | **49.25±1.48** | 415 | 41.11±1.40 | 397 |
| 128 | **50.60±1.37** | 421 | **49.92±1.47** | 421 | 47.46±1.45 | 412 |
| 256 | **51.12±1.46** | 421 | **50.90±1.49** | 422 | **49.75±1.51** | 417 |
| SDPA (baseline) | | | | | 49.86±1.47 | 424 |

We further evaluate SANTA on the MMLU benchmark (Hendrycks et al., 2021; Center for AI Safety, 2024), which tests factual recall and general reasoning. MMLU benchmarking observes similar trends as GSM8K, where SANTA virtually recovers baseline accuracy at $S = 256$, which is significantly shorter than the average sequence length $n_k = 417$. $S^2$ANTA variants outperform default SANTA across $S$ budgets of 64–256, recovering baseline SDPA accuracy (within 1%) with multiplier-free arithmetic and sparse memory access to rows of $V$ since $S \ll n_k$. Extended results with DeepSeek 7B and top-k comparisons are provided in Appendix H.

### 4.3. Long-Context Benchmarks

We benchmark SANTA on long-context tasks borrowed from RULER (Hsieh et al., 2024) with a prompt length of 8192 tokens. The theoretical benefit of $S^2$ANTA's sparse computation is most stark in this long-context regime. With a budget of $S = 256$ in an 8192-token sequence, **$S^2$ANTA**

*Table 6.* Accuracy of Llama 8B on 4 long-context tasks with an 8k-token prompt. Tasks: frequent word extraction (FWE), needle-in-a-haystack (NIAH), single-hop QA ($QA_1$), multi-hop QA ($QA_2$). $S$: SANTA sample budget. Accuracy shows 95% bootstrap confidence intervals.

| Kernel | $S$ | FWE | NIAH | $QA_1$ | $QA_2$ |
|---|---|---|---|---|---|
| $S^2$ANTA-sys | 64 | 96.27±0.97 | 94.05±1.07 | 71.80±3.90 | 67.20±4.20 |
| $S^2$ANTA-sys | 128 | 97.80±0.70 | 94.15±1.02 | 68.20±4.10 | 66.80±4.00 |
| **$S^2$ANTA-sys** | **256** | **97.87±0.70** | **95.00±0.95** | **71.20±3.90** | **69.40±4.10** |
| $S^2$ANTA-strat | 64 | 95.93±0.94 | 94.35±1.15 | 71.40±4.00 | 65.40±4.20 |
| $S^2$ANTA-strat | 128 | 97.73±0.80 | 94.30±1.10 | 70.80±4.20 | 67.40±4.20 |
| **$S^2$ANTA-strat** | **256** | **98.33±0.60** | **94.55±1.05** | **71.20±4.00** | **67.00±4.20** |
| SANTA | 64 | 92.53±1.27 | 93.95±1.30 | 64.60±4.20 | 63.40±4.20 |
| SANTA | 128 | 94.53±1.13 | 91.45±1.38 | 67.80±4.10 | 63.40±4.30 |
| SANTA | 256 | 96.20±0.93 | 93.15±1.18 | 68.00±4.10 | 66.40±4.10 |
| **SDPA (baseline)** | **—** | **98.53±0.60** | **94.55±1.00** | **71.80±3.90** | **68.60±4.20** |

**uses just $S/n_k = 3.125\%$ of the value-stage additions and memory accesses compared to full attention** (following Table 10). It achieves this while using **no multiplications** and recovering baseline SDPA accuracy (within $\pm1\%$) across every task.

For full clarity, when Llama 8B's grouped-query attention (Ainslie et al., 2023) with 4 grouped queries is fully accounted for, then memory access to unique rows of $V$ can in the worst case reach 12.5%, which remains extremely sparse. Extended results with top-k comparisons are provided in Appendix H.

## 5. Bernoulli $qK^\top$ for Sparse Key Access

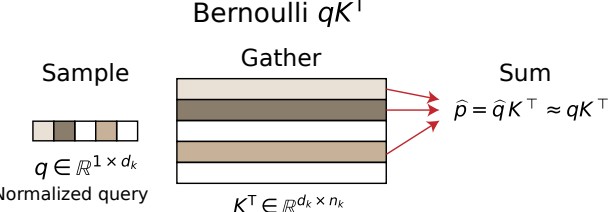

*Figure 5.* Bernoulli $qK^\top$ represents elements of the query vector as Bernoulli probabilities, sparsifying along the feature dimension $d_k$ with multiplier-free arithmetic.

SANTA reduces memory access for the value matrix by sampling from the softmax distribution; however, it leaves the score computation ($qK^\top$) intact. As illustrated in Fig. 5, we demonstrate that Bernoulli sampling of the query naturally zeroes out query-irrelevant features, thereby effectively reducing memory access for the key matrix.

We propose representing each query element as an independent Bernoulli variable, allowing the corresponding key feature to be pruned when the sampling probability is low. While this method is applicable to the prefill phase, it in-

duces sparse key access specifically during decoding; therefore, we focus on a single query $q$. First, the query vector is normalized as $q \leftarrow q/\text{norm}$, where $\text{norm} \geq \max_i |q_i|$. This ensures that each element $q_i \in [-1, 1]$ can be interpreted as a probability. Next, we estimate the product $p = qK^\mathsf{T}$ as:

$$\widehat{p} = \frac{\text{norm}}{B} \sum_{n=1}^{B} \widehat{q}^{(n)} K^\mathsf{T}, \qquad (5)$$

where $B$ is the sample count and $\widehat{q}^{(n)}$ is a sparse ternary vector constructed via Bernoulli sampling (detailed in Appendix C). The linearity of expectation ensures this estimator is unbiased. Because $\widehat{q}^{(n)}$ contains only values in $\{-1, 0, +1\}$, $\widehat{p}$ can be computed without multiplications by gathering and accumulating selected rows of $K^\mathsf{T}$, followed by normalizing bit shifts for fixed-point representations.

Table 7 shows GSM8K results for Bernoulli $qK^\mathsf{T}$ applied to BitNet 2B (Ma et al., 2025) for both prefill and decoding. Using $B = 4$ stratified samples nearly recovers the accuracy of standard SDPA for the GSM8K benchmark, although the approximated scores typically have a 30% error (Appendix C.1). This suggests that the $qK^\mathsf{T}$ product within the highly quantized BitNet 2B architecture can tolerate numerical errors while maintaining strong overall performance.

Crucially, Bernoulli $qK^\mathsf{T}$ enables sparse access to cached keys, a benefit that is uniquely effective during decoding (the prefill phase requires nearly all features). This is shown in Table 7: with $B = 4$, the model needs only 67.5% of key features per head (vs. $\sim 100\%$ for prefill), while maintaining accuracy within 2% of the SDPA baseline.

*Table 7.* Performance of Bernoulli $qK^\mathsf{T}$ (stratified) on GSM8K (BitNet 2B) with the test set evaluated three times, and confidence intervals obtained by bootstrapping.

| Method | B | Acc. (%) (95% CI) | Avg. Tok. | $K^\mathsf{T}$ Access (%) Head | Group |
|---|---|---|---|---|---|
| **SDPA** | – | $65.7 \pm 1.5$ | 306.1 | 100 | 100 |
| | 1 | $1.9 \pm 0.43$ | 1478.6 | 25.6 | 65.8 |
| | 2 | $48.7 \pm 1.4$ | 345.9 | 46.3 | 88.0 |
| Bernoulli $qK^\mathsf{T}$ | 4 | $\mathbf{64.5 \pm 1.5}$ | 309.4 | **67.5** | 97.9 |
| | 8 | $65.1 \pm 1.5$ | 316.9 | 82.6 | 99.8 |
| | 16 | $65.8 \pm 1.5$ | 317.7 | 91.1 | 100 |
| | 2 | $48.6 \pm 1.5$ | 318.6 | 57.9 | 57.9 |
| Mean Group Query | 4 | $63.8 \pm 1.5$ | 311.4 | **84.7** | **84.7** |
| Bernoulli $qK^\mathsf{T}$ | 8 | $65.6 \pm 1.6$ | 311.4 | 97.1 | 97.1 |
| | 16 | $66.2 \pm 1.5$ | 310.2 | 99.7 | 99.7 |

### 5.1. Mean Group Query

While Bernoulli $qK^\mathsf{T}$ effectively prunes irrelevant features for individual heads, Grouped Query Attention (GQA) (Ainslie et al., 2023) complicates this by sharing keys and values across multiple heads (e.g., groups of 4 in BitNet).

Consequently, the set of accessed features becomes the union of those required by all queries in the group, significantly increasing memory traffic. Table 7 illustrates this effect: with $B = 4$ samples, $K^\mathsf{T}$ access jumps from 67.5% per head to 97.9% for the group. To preserve sparsity in this scenario, we propose using a representative query $m$ for the entire group, defined as the mean of the queries' absolute values. We then construct the estimator $\widehat{m}$ for the mean query $m$ via Monte Carlo sampling (detailed in Appendix C.2), which stochastically prunes the low-amplitude mean features.

With $B = 4$ samples, $\widehat{m}$ contains approximately 15% of zero elements, which allows fetching only 85% of key features for the group (Table 7). The product $p = qK^\mathsf{T}$ for each query is finally estimated as:

$$\widehat{p} = \left( \widehat{m} \odot \frac{q}{m} \right) K^\mathsf{T}, \qquad (6)$$

where $\odot$ denotes element-wise multiplication. In this scenario, calculating $\widehat{p}$ requires normalization by $m$ to compensate for the bias introduced by the mean group query. As shown in Table 7, the resulting estimator achieves similar accuracy to the single-query configuration while saving 15% of key memory accesses. We employ the mean group query estimator (Eq. 6) in all the following experiments.

### 5.2. Long-Context Benchmarks

We evaluate our method on long-context tasks using Llama 8B, applying Bernoulli $qK^\mathsf{T}$ strictly during decoding since prefill requires full feature access. As detailed in Table 8, using $B = 8$ samples reduces key feature access to 72.8% for Llama 8B, with minimal accuracy loss ($< 2\%$ on NIAH, $QA_1$, and $QA_2$). Furthermore, this experiment highlights that key memory access patterns vary significantly across models. In contrast to Llama 8B, BitNet 2B requires accessing 97.1% of features for the same budget ($B = 8$, Table 7), a difference we attribute to distinct query distributions.

*Table 8.* Accuracy of Llama 8B on 4 long-context tasks with a 4,096-token prompt. Tasks: frequent word extraction (FWE), needle-in-a-haystack (NIAH), single-hop QA ($QA_1$), multi-hop QA ($QA_2$). $B$: Bernoulli $qK^\mathsf{T}$ (mean group query) sample budget. Accuracy shows 95% bootstrap confidence intervals.

| Kernel ($qK^\mathsf{T}$) | B | $K^\mathsf{T}$ Access (%) | FWE | NIAH | $QA_1$ | $QA_2$ |
|---|---|---|---|---|---|---|
| Bernoulli | 4 | 47.3 | $70.4 \pm 2.0$ | $87.2 \pm 1.9$ | $72.0 \pm 3.8$ | $66.2 \pm 4.1$ |
| Bernoulli | **8** | **72.8** | $89.4 \pm 1.4$ | $\mathbf{93.8 \pm 1.4}$ | $\mathbf{73.6 \pm 3.9}$ | $\mathbf{70.2 \pm 3.9}$ |
| Bernoulli | 16 | 92.1 | $92.7 \pm 1.2$ | $94.5 \pm 1.2$ | $74.8 \pm 3.5$ | $71.6 \pm 4.2$ |
| **SDPA** | – | – | $\mathbf{94.0 \pm 1.1}$ | $\mathbf{95.0 \pm 1.2}$ | $\mathbf{73.8 \pm 3.8}$ | $\mathbf{70.8 \pm 4.0}$ |

### 5.3. Combining Bernoulli $qK^\mathsf{T}$ and SANTA

Combining SANTA with Bernoulli $qK^\mathsf{T}$ enables sparse key–value accesses during decoding while replacing the majority

of multiplications with additions. Table 9 shows the accuracy for the GSM8K benchmark using BitNet 2B and $B = 4$ Bernoulli samples for the $qK^\top$ computation, as the number of SANTA samples $S$ is varied for the value stage. Using $B = 4$ for queries and $S = 64$ for attention weights almost recovers the SDPA baseline with less than 2% accuracy drop. Critically, this combined approach reduces key access by 15.3% and value access by 89.2%.

*Table 9.* Performance of Bernoulli $qK^\top$ (mean group query) + SANTA on GSM8K (BitNet 2B) with the test set evaluated three times (CI computed by bootstrapping).

| Config | Acc. (%) | Avg. | (K,V) Access (%) |
|---|---|---|---|
| | (95% CI) | Tok. | Group |
| **SDPA** | **65.7 ± 1.5** | 306.1 | (100, 100) |
| $B = 4, S = 8$ | 55.0 ± 1.6 | 319.2 | (84.7, 2.4) |
| $B = 4, S = 16$ | 60.7 ± 1.5 | 313.3 | (84.7, 3.7) |
| $B = 4, S = 32$ | 62.3 ± 1.5 | 308.4 | (84.7, 6.2) |
| **B = 4, S = 64** | **63.8 ± 1.5** | 304.6 | **(84.7, 10.8)** |
| $B = 4, S = 128$ | 64.1 ± 1.5 | 303.2 | (84.7, 14.3) |

# 6. Related Work

**Exact attention kernels and memory-bound decoding.** FlashAttention-style kernels optimize IO locality and utilization for *exact* attention, including decoding-focused variants (Dao et al., 2023; Ye et al., 2025). However, exact decoding still streams essentially the full KV state each step at long contexts. Our approach is complementary: rather than further optimizing a full KV scan, we reduce the amount of KV data accessed at runtime by sparsifying value-cache reads (and, for Bernoulli $qK^\top$, key-feature reads).

**Reducing KV footprint, retention, and selection.** KV quantization/compression reduces bytes per KV element (Liu et al., 2024; Hooper et al., 2024), weight-only quantization targets model parameters (Frantar et al., 2023; Dettmers et al., 2023), and GQA reduces KV size through head sharing (Ainslie et al., 2023). Recent long-context systems also reduce the effective cache through selection, retention, or offloading: ShadowKV (Sun et al., 2025), OmniKV (Hao et al., 2025), and DuoAttention (Xiao et al., 2025) decide which KV states are retained, reconstructed, selected, or assigned to retrieval/streaming heads. These methods answer a different systems question from SANTA. They reduce or restructure the candidate KV state before attention is evaluated, whereas SANTA modifies the softmax–$V$ aggregation over whatever KV state is presented to the operator. Thus, after an upstream method reduces the candidate set from $n$ to $n'$, SANTA can in principle replace dense aggregation over $n'$ value rows with sampled aggregation over $S$ rows. The marginal benefit will depend on $n'$, hardware, batching, and task, so we view such hybrids as complementary but leave empirical evaluation to future

work. MLA-style architectures (DeepSeek-AI et al., 2024) may similarly reduce the KV bandwidth share targeted by SANTA, making them an important future benchmark.

**Approximating attention and sampling vs. truncation.** Structured sparsity and low-rank approximations reduce attention cost (often in prefill) (Child et al., 2019; Zaheer et al., 2020; Beltagy et al., 2020; Wang et al., 2020). In decoding, candidate selection and query-pruning methods reduce the cost of identifying relevant tokens or approximating $qK^\top$ (Kitaev et al., 2020; Chen et al., 2025; Ribar et al., 2024). This is where our two methods differ in scope: **SANTA operates in the value stage and is orthogonal to score-stage approximations**, while **Bernoulli $qK^\top$ is an alternative score-stage estimator** that induces feature-sparse key access via stochastic ternary queries.

Sampling-based attention approximations have also been explored (Kim & Ko, 2022). A direct deterministic baseline is top-$k$ attention (Gupta et al., 2021), which reduces value reads by truncation but is biased. In contrast, SANTA provides an unbiased estimator of the post-softmax value aggregation with controllable variance, and we emphasize variance reduction (stratified/systematic sampling) together with GPU kernels that realize measured speedups in the memory-bound decoding regime. Related stochastic-computing implementations in spiking Transformer architectures are orthogonal to this focus (Song et al., 2024). We do not claim that stochastic sampling dominates deterministic truncation in all regimes: Appendix H shows that top-$k$ is competitive on short-context GSM8K. In the long-context regime targeted by SANTA, however, the same appendix shows $S^2$ANTA outperforming top-$k$ across nearly all 8k-token tasks at matched budgets, especially on multi-hop QA, where the answer may depend on several parts of the context rather than only the tokens with the largest attention scores.

# 7. Conclusion

We introduced SANTA, a stochastic sparse-attention method for long-context autoregressive decoding that samples $S \ll n_k$ value rows from the post-softmax distribution and aggregates them with gather-and-add. The estimator is unbiased, admits practical variance reduction through stratified/systematic sampling, and gives a tunable accuracy–latency knob through the sample budget $S$. On Llama-3.1-8B-Instruct, our $S^2$ANTA kernels match baseline accuracy on 32k-token RULER tasks and LongBench v2 while achieving up to $1.5\times$ decode-step attention-kernel speedup and up to $1.25\times$ end-to-end decode-latency speedup on RTX 6000 Ada. SANTA is specialized for long-context, decode-heavy workloads and is complementary to cache compression, retention, offloading, and selection methods. We also presented Bernoulli $qK^\top$ sampling as a forward-looking score-stage mechanism for sparse key-feature access.

## Acknowledgments

Research was sponsored by the Army Research Office and was accomplished under Grant Number W911NF-24-1-0228. The views and conclusions contained in this document are those of the authors and should not be interpreted as representing the official policies, either expressed or implied, of the Army Research Office or the U.S. Government. The U.S. Government is authorized to reproduce and distribute reprints for Government purposes notwithstanding any copyright notation herein.

This material is based upon work supported by the National Science Foundation Graduate Research Fellowship Program under Grant No. 2139319. Any opinions, findings, and conclusions or recommendations expressed in this material are those of the author(s) and do not necessarily reflect the views of the National Science Foundation.

This work used resources available through the National Research Platform (NRP) at the University of California, San Diego. NRP has been developed, and is supported in part, by funding from National Science Foundation, from awards 1730158, 1540112, 1541349, 1826967, 2112167, 2100237, and 2120019, as well as additional funding from community partners.

## Impact Statement

This paper presents work whose goal is to advance the field of machine learning. There are many potential societal consequences of our work, none of which we feel must be specifically highlighted here.

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

# A. Formal Description of SANTA

For query $q$, the SANTA estimator can be written as

$$\widehat{AV}_q = \frac{1}{S} \sum_{s=1}^{S} V_{i_{q,s}}, \qquad i_{q,s} \sim \text{Categorical}(A_q). \quad (7)$$

**Definition A.1** (SANTA estimator). Let $Q \in \mathbb{R}^{n_q \times d_k}$, $K \in \mathbb{R}^{n_k \times d_k}$, $A = \text{softmax}\big(QK^\mathsf{T}/\sqrt{d_k}\big) \in \mathbb{R}^{n_q \times n_k}$, and $V \in \mathbb{R}^{n_k \times d_k}$. For each query index $r \in \{1, \ldots, n_q\}$, sample $i_{r,1}, \ldots, i_{r,S} \overset{\text{i.i.d.}}{\sim} \text{Categorical}(A_r)$ and define $\widehat{AV}_r = \frac{1}{S} \sum_{s=1}^{S} V_{i_{r,s}}$. Stacking all queries gives $\widehat{AV} \in \mathbb{R}^{n_q \times d_k}$.

## A.1. Unbiasedness of SANTA

**Proposition A.2** (Unbiasedness). $\mathbb{E}[\widehat{AV}] = AV$.

*Proof of Proposition A.2.* Fix any $q$. Since $\Pr(i_{q,s} = j) = A_{qj}$,

$$\mathbb{E}[\widehat{AV}_q] = \frac{1}{S} \sum_{s=1}^{S} \sum_{j=1}^{n_k} A_{qj} V_j = \sum_{j=1}^{n_k} A_{qj} V_j = (AV)_q.$$

Linearity of expectation completes the result. □

## A.2. Variance of SANTA

**Proposition A.3** (Variance scaling of SANTA). *The covariance of the SANTA estimator scales as $1/S$.*

Throughout this subsection, fix a query index $q \in \{1, \ldots, n_q\}$. Let $A_q \in \Delta^{n_k-1}$ denote the attention weights over keys, write $p_{qj} \triangleq A_{qj}$, and let $\mu_q \triangleq \sum_{j=1}^{n_k} p_{qj} V_j = (AV)_q$ be the exact attention output for that query. Default SANTA draws $i_{q,1}, \ldots, i_{q,S} \overset{\text{i.i.d.}}{\sim} \text{Categorical}(A_q)$ and returns

$$\widehat{AV}_q = \frac{1}{S} \sum_{s=1}^{S} V_{i_{q,s}} \in \mathbb{R}^{d_k}.$$

Define centered summands $X_s \triangleq V_{i_{q,s}} - \mu_q$ and the (query-specific) covariance $\Sigma_q \triangleq \text{Cov}(V_i) = \sum_{j=1}^{n_k} p_{qj} (V_j - \mu_q)(V_j - \mu_q)^\mathsf{T}$.

We now aim to show that SANTA's covariance scales as $1/S$ (Proposition A.3):

$$\mathbb{E}[\widehat{AV}_q] = \mu_q, \qquad \text{Cov}(\widehat{AV}_q) = \frac{1}{S} \Sigma_q.$$

Equivalently, $\mathbb{E}\left[\|\widehat{AV}_q - \mu_q\|_2^2\right] = \frac{1}{S} \text{tr}(\Sigma_q)$.

*Proof.* Unbiasedness was shown in Proposition A.2. For the variance, write $\widehat{AV}_q - \mu_q = \frac{1}{S} \sum_{s=1}^{S} X_s$ with $\mathbb{E}[X_s] = 0$ and $\text{Cov}(X_s) = \Sigma_q$. The $X_s$ are i.i.d., hence $\text{Cov}\left(\frac{1}{S} \sum_{s=1}^{S} X_s\right) = \frac{1}{S^2} \sum_{s=1}^{S} \text{Cov}(X_s) = \frac{1}{S} \Sigma_q$. Taking the trace gives the mean-squared error identity. □

## A.3. $S^2$ANTA Estimator

**Proposition A.4** (Unbiasedness of $S^2$ANTA). *Let $\mu_q := \sum_j p_{qj} V_j = (AV)_q$. Then $\mathbb{E}[\widehat{AV}_q^{\text{ind}}] = \mathbb{E}[\widehat{AV}_q^{\text{sys}}] = \mu_q$.*

The proof is shown in Section A.4.

*Remark A.5* (Systematic vs. independent). $S^2$ANTA-strat is unbiased and admits a variance-dominance theorem, guaranteeing lower variance than that of i.i.d. sampling with default SANTA (Section A.5). $S^2$ANTA-sys is also unbiased, but does not provide tractable variance guarantees; empirically it performs comparably.

## A.4. Unbiasedness of $S^2$ANTA

*Proof of Proposition A.4.* For $S^2$ANTA-strat,

$$
\begin{aligned}
\mathbb{E}[\widehat{AV}_q^{\text{ind}}] &= \frac{1}{S} \sum_{m=0}^{S-1} \mathbb{E}\Big[V_{F_q^{-1}(T_m)}\Big] \\
&= \int_0^1 \Big( \sum_j \mathbf{1}\{F_q(j-1) \le t < F_q(j)\} V_j \Big)\, dt \\
&= \sum_j p_{qj} V_j \\
&= \mu_q.
\end{aligned}
$$

For $S^2$ANTA-sys, observe that *marginally* each $T_m$ is uniform on $I_m$ (though dependent across $m$). By linearity of expectation, the sum of expectations is unchanged by this dependence, so the same integral argument applies; see Cochran (1977, Ch. 8) and Owen (2013). □

## A.5. Variance Reduction for $S^2$ANTA-Strat

**Theorem A.6** (Variance reduction for $S^2$ANTA-strat). *Let* $\Sigma_q := \sum_j p_{qj}(V_j - \mu_q)(V_j - \mu_q)^\top$ *and define within-stratum covariances*

$$
\Sigma_q^{(m)} := \text{Cov}\Big(V_{F_q^{-1}(T)} \,\Big|\, T \sim \text{Unif}(I_m)\Big).
$$

*Then*

$$
\text{Cov}(\widehat{AV}_q^{\text{ind}}) = \frac{1}{S^2} \sum_{m=0}^{S-1} \Sigma_q^{(m)} \preceq_{\text{Loewner}} \frac{1}{S} \Sigma_q = \text{Cov}(\widehat{AV}_q),
$$

*with strict improvement unless all stratum means coincide. See Lohr (2010, Ch. 4) and Cochran (1977, Ch. 5).*

*Proof.* Independence across strata gives $\text{Cov}(\widehat{AV}_q^{\text{ind}}) = \frac{1}{S^2} \sum_m \text{Cov}\Big(V_{F_q^{-1}(T)} \mid T \in I_m\Big)$. By the law of total covariance with $T \sim \text{Unif}(0,1)$ and partition $\{I_m\}$,

$$
\begin{aligned}
\Sigma_q &= \mathbb{E}\Big[\text{Cov}\Big(V_{F_q^{-1}(T)} \mid T \in I_m\Big)\Big] \\
&\quad + \text{Cov}\Big(\mathbb{E}\Big[V_{F_q^{-1}(T)} \mid T \in I_m\Big]\Big) \\
&\succeq_{\text{Loewner}} \mathbb{E}\Big[\text{Cov}\Big(V_{F_q^{-1}(T)} \mid T \in I_m\Big)\Big].
\end{aligned}
$$

and averaging then dividing by $S$ yields the claim. Strictness holds unless the stratum means are all equal. □

**Corollary A.7** (MSE ordering). *For any $q$,*
$$
\mathbb{E}\Big[\|\widehat{AV}_q^{\text{ind}} - \mu_q\|_2^2\Big] \le \mathbb{E}\Big[\|\widehat{AV}_q - \mu_q\|_2^2\Big] = \frac{1}{S}\text{tr}(\Sigma_q).
$$

## B. Vector Bernstein Tail for SANTA

**Theorem B.1** (Vector Bernstein tail for SANTA). *Assume a uniform almost-sure bound $\|V_j - \mu_q\|_2 \le V_{\max}$ for all $j$ (hence $\|X_s\|_2 \le V_{\max}$). Let $v_q \triangleq \mathbb{E}\big[\|V_i - \mu_q\|_2^2\big] = \text{tr}(\Sigma_q)$. Then for all $t > 0$,*

$$
\Pr\Big(\|\widehat{AV}_q - \mu_q\|_2 \ge t\Big) \le 2\exp\Big(-\frac{S\,t^2}{2\,(v_q + V_{\max}t/3)}\Big).
$$

*Equivalently, with probability at least $1 - \delta$,*

$$
\|\widehat{AV}_q - \mu_q\|_2 \le \sqrt{\frac{2\,v_q\,\log(2/\delta)}{S}} + \frac{2V_{\max}\,\log(2/\delta)}{3S}.
$$

*Proof sketch.* This is a Hilbert-space (vector) Bernstein inequality: apply Bernstein to the mean of independent, mean-zero, $L$-bounded vectors with variance proxy $v_q$; see, e.g., Boucheron et al. (2013, Thm. 2.10; see also Cor. 2.11) and Pinelis (1994). □

## C. Estimating $qK^\top$ via Bernoulli Sampling

We first normalize the query vector as $q \leftarrow q/\text{norm}$, where $\text{norm} \ge \max_i |q_i|$ so that each element $q_i \in [-1, 1]$ is interpreted as a probability. Then, we introduce the unbiased query estimator $\widehat{q}$ where each element $\widehat{q}_i$ is an independent ternary Bernoulli variable:

$$
\begin{aligned}
b_i &\sim \text{Bernoulli}(|q_i|), \\
\widehat{q}_i &= b_i \times \text{sign}(q_i) \in \{-1, 0, +1\}.
\end{aligned} \tag{8}
$$

Sampling from $\widehat{q}$ provides an unbiased estimator for the product $p = qK^\top$:

$$
\widehat{p} = \frac{\text{norm}}{B} \sum_{n=1}^B \widehat{q}^{(n)} K^\top, \tag{9}
$$

where $\widehat{q}^{(n)}$ is a Bernoulli query vector, and $B$ is the number of samples.

### C.1. Bernoulli $qK^\top$ Error

Fig. 6 shows the scaling of the L2 norm error with the number of samples $B$, defined as $e = \|\widehat{p} - p\|_2 / \|p\|_2$. In similar fashion as SANTA, stratified Bernoulli sampling reduces the variance of each independent element $\widehat{p}_i$. The variance scales as $O(1/B^2)$, resulting in an overall L2 norm scaling as $O(1/B)$. For this numerical experiment, which uses a head dimension $d_k = 128$ and a context length $n_k = 1024$, using only $B = 4$ samples induces a mean L2 norm error of approximately $60\%$ for standard Bernoulli sampling, compared to a much lower $30\%$ for stratified sampling.

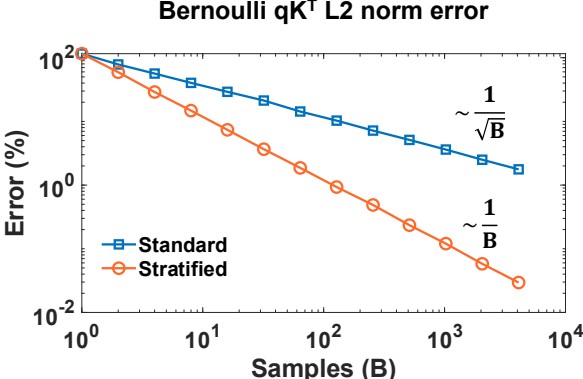

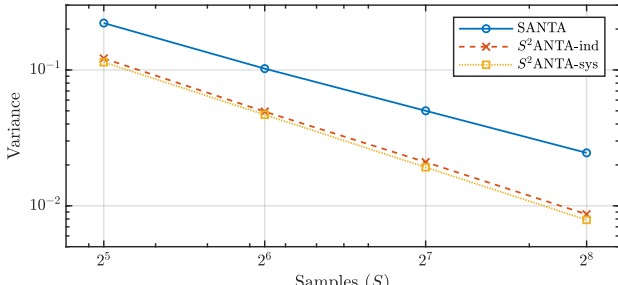

*Figure 7.* Empirical variance of SANTA on single-hop QA prompts (8k tokens).

*Figure 6.* Mean L2 norm error for Bernoulli $qK^T$. The score error is calculated during decoding with $d_k = 128$, a context length $n_k = 1024$, and averaged over 100 instances $q \sim \mathcal{N}(0,1)$, $K \sim \mathcal{N}(0,1)/\sqrt{d_k}$, $V \sim \mathcal{N}(0,1)$.

## C.2. $qK^\mathsf{T}$ for Grouped Query Attention

To maintain sparse key access when keys share multiple queries, as in GQA, we define a common representative query for the group, such as the mean of the absolute values of the queries:

$$m = \frac{1}{|\mathcal{G}|} \sum_{g \in \mathcal{G}} |q_g|, \qquad (10)$$

where $\mathcal{G}$ denotes the query group. We then estimate the mean query using Monte Carlo sampling, after normalizing by norm $\geq \max_i m_i$:

$$b_i \sim \text{Bernoulli}(m_i/\text{norm}) \qquad (11)$$

$$\widehat{m} = \frac{\text{norm}}{B} \sum_{n=1}^{B} b^{(n)} \qquad (12)$$

where $b^{(n)}$ are Bernoulli query vectors with independent elements $b_i$. After fetching the corresponding $K^\mathsf{T}$ rows corresponding to non-zero elements of $\widehat{m}$, the product $p = qK^\mathsf{T}$ for each query is estimated as:

$$\widehat{p} = \left(\widehat{m} \odot \frac{q}{m}\right)K^\mathsf{T}, \qquad (13)$$

where $\odot$ denotes element-wise multiplication. In this mean group query scenario, calculating $\widehat{p}$ requires multiplications by the original query $q$ and normalization by $m$ to compensate for the bias introduced by the mean query approximation.

## D. Empirical Variance

To further validate our results, we empirically measure the variance of SANTA, $S^2$ANTA-strat, and $S^2$ANTA-sys. We sweep the sample budget $S$ on 8k-token single-hop QA prompts, averaged across model layers and 30 prompts (normalized per head). $S^2$ANTA variants exhibit significantly lower variance than SANTA. Though systematic sampling admits no tractable theoretical variance guarantees, it empirically demonstrates similar variance to independent stratified sampling (and requires 1 random number per query instead of $S$ random numbers). On a log-log scale, slopes for SANTA, $S^2$ANTA-strat, and $S^2$ANTA-sys are $-1.1$, $-1.27$, and $-1.29$, respectively, matching the $1/S$ scaling prescribed by theory (see Appendix N for details regarding the measurement protocol).

## E. Theoretical Operations and Memory Accesses in Autoregressive Decoding

*Table 10.* Per-query, per-head value-stage compute and memory for dense scaled dot-product attention (SDPA) and SANTA. $n_k$ is the number of keys, $d_k$ is the head dimension, and $S$ is the SANTA sample budget per query. Value reads counts $V$ elements accessed per query; Writes counts output elements.

| Metric | SDPA | SANTA |
|---|---|---|
| Adds | $n_k d_k$ | $S d_k$ |
| Mults / Divs | $n_k d_k$ | $d_k$ |
| Value reads | $n_k d_k$ | $S d_k$ |
| Writes | $d_k$ | $d_k$ |

Table 10 displays the theoretical additions, multiplications, and memory accesses incurred by SANTA in the value stage of attention, where the attention scores $A$ multiply $V$. The score computation $qK^\mathsf{T}$ is similar for both SANTA and SDPA, and is therefore omitted in this analysis (see Appendix G). We count individual scalar elements which are added together, multiplied together, or accessed by memory. Architecture-dependent effects such as multiply-accumulate (MAC) instructions, caching effects, or operation parallelism are not accounted for; arguments are fundamental and hardware-agnostic.

By virtue of SANTA's sparsity, SANTA demonstrates a $S/n_k$ reduction in post-softmax additions and reads to rows

of $V$, where $S$ is the sample budget and $n_k$ is the number of keys (sequence length). A key distinction is that SANTA costs *fewer* additions than SDPA; SANTA does not seek to replace the multiplications in the $AV$ computation, but rather, leverages sparse sampling to decrease the total number of additions while eliminating multiplications. So long as $S \ll n_k$, the advantage of attention sparsity holds true. Sampling overhead is not included; it is highly dependent on implementation, but drawing $S \ll n_k$ samples should be lightweight relative to memory accesses, multiplies, and adds (indeed, this proves to be the case with GPU kernel latency demonstrations in Section 3).

Whereas SDPA incurs $n_k d_k$ multiplications from computing dot products, SANTA only costs $d_k$ normalizing multiplies by $1/S$. $d_k$ is a constant model parameter and does not increase with the sequence length $n_k$. Furthermore, if $S$ is chosen as a power of 2 and the model has a fixed point representation, this normalization may be implemented as a bit-shift.

In prefill, while similar arguments hold for SANTA's reduced additions and multiplies, there is no longer sparse memory access to the value matrix $V$. Each of $n_q$ queries may each sample $S$ rows of $V$, therefore requiring $\gg S$ rows of $V$ when the union across all queries is considered (see Appendices F, G).

## F. Unique $V$-Row Coverage for SANTA Prefill

*Table 11.* Unique $V$ rows read (union over all query positions in a causal prefill pass), reported as a percentage of all $n_k$ rows in the value matrix. Each entry is the mean over 10 trials and all attention heads, with Gaussian inputs $Q, K, V \sim \mathcal{N}(0, 1)$ (FP16), $H{=}32$ heads, and head dimension $d_k{=}128$.

| $S$ | Unique $V$ rows read (% of $n_k$) | | | | |
|---|---|---|---|---|---|
| | $n_k{=}1024$ | 2048 | 4096 | 8192 | 16384 |
| 1 | 49.90 | 50.04 | 49.96 | 49.95 | 49.95 |
| 4 | 79.95 | 80.00 | 79.96 | 79.96 | 79.95 |
| 8 | 88.92 | 88.83 | 88.86 | 88.86 | 88.85 |
| 16 | 94.07 | 94.11 | 94.10 | 94.09 | 94.08 |

Table 11 shows the percentage of unique $V$ rows read by SANTA during prefill passes of differing sequence lengths $n_k$. Standard Gaussian distributed inputs $Q, K, V$ are passed into SANTA with $d_k = 128$ and $H = 32$ heads for multi-headed attention. Even in the long-context, $S \ll n_k$ regime, any appreciable number of samples $S$ results in SANTA sampling most rows of $V$ when the union across all queries is considered. For $S = 16$, selected sequence lengths from $n_k = 1k - 16k$ show SANTA reading almost all ($> 94\%$) rows of $V$, because each of $n_q = n_k$ queries stochastically attend to $S$ sampled keys. Therefore, while SANTA features sparse $V$ access during autoregressive decoding, SANTA does *not* have apparent benefit from sparse

memory access in prefill. Arguments concerning SANTA's reduced addition and multiplication operations are still valid in prefill (Table 12).

## G. Theoretical Operations and Memory Accesses in Prefill

*Table 12.* Per-head FLOPs and memory for scaled dot-product attention (SDPA), top-$k$, and SANTA. $n_q$: number of queries, $n_k$: number of keys, $d_k$: head dimension, $k$: top-$k$ budget, $S$: SANTA sample budget per query. "Reads" count logical value/key elements accessed; "Writes" count output or score elements. Softmax, top-$k$ selection, and sampling overheads are omitted.

| Stage / Variant | Adds | Mults / Divs | Reads | Writes |
|---|---|---|---|---|
| **Score stage ($QK^\mathsf{T}$)** | | | | |
| SDPA / top-$k$ / SANTA | $n_q n_k d_k$ | $n_q n_k d_k$ | $(n_q + n_k)d_k$ | $n_q n_k$ |
| **Value stage** | | | | |
| SDPA | $n_q n_k d_k$ | $n_q n_k d_k$ | $n_q n_k d_k$ | $n_q d_k$ |
| Top-$k$ | $n_q k d_k$ | $n_q k d_k$ | $n_q k d_k$ | $n_q d_k$ |
| SANTA | $n_q S d_k$ | $n_q d_k$ | $n_q S d_k$ | $n_q d_k$ |

While SANTA's apparent memory sparsity benefits are most viable in autoregressive decoding, for full clarity, this section provides generalized analysis for prefill. We consider the computational cost of SANTA, SDPA, and top-k attention, focusing on FLOPs and memory traffic in the value stage. Table 12 summarizes the theoretical costs per attention head. For the score stage ($QK^\mathsf{T}$), the "Reads" entry $(n_q + n_k)d_k$ counts each query and key vector once per head, assuming they are loaded once from DRAM and then reused in on-chip memory during the matmul. In contrast, for the value stage we report $n_q n_k d_k$, $n_q k d_k$, and $n_q S d_k$ for dense SDPA, top-$k$, and SANTA, respectively, which count the per-query accesses rather than unique loads of elements from memory.

While top-$k$ attention is memory-efficient and generally performs well, it introduces bias and can degrade significantly when the attention distribution is heavy-tailed. For example, recent work (Chen et al., 2025) shows that tasks such as common word extraction suffer when relevant information is not concentrated in the top-$k$ entries. In contrast, stochastic estimators like SANTA provide an unbiased estimate of full attention and retain robustness in such cases.

For both top-k and SANTA, we define a notion of compute budget, where $k$ is the number of keys selected by top-k for each query of each attention head, while $S$ is a sample budget representing the number of samples that SANTA draws for each query of each attention head. By default, we employ uniform $S$ across model layers, though ablations suggest that $S$ can be optimized per layer (Appendix J, K).

We omit softmax computation, top-$k$ sorting overhead, and sampling from this analysis, as they are lightweight relative to memory accesses and the $V$ matrix multiply, and are

highly implementation-dependent. For example, many implementations use partial sorting (Samaga B L et al., 2024; Xie et al., 2025).

In both FLOPs and memory reads, SDPA scales as $n_q n_k$, reflecting the cost of full quadratic attention computation. Both top-$k$ and SANTA reduce this to $n_q k$ and $n_q S$, respectively, assuming fixed budgets with $k, S \ll n_k$. This shifts the overall complexity of the value stage from quadratic to linear in sequence length, given a fixed budget $k$ or $S$.

Top-$k$ still incurs multiplication costs while SANTA eliminates multiplies. Top-$k$ performs $n_q k d_k$ multiplications (per head), while SANTA requires only $n_q d_k$ divisions for normalization (if $S$ is chosen as a power of 2, even these few divisions become simple bit-shifts).

# H. Additional DeepSeek and Top-k Results

*Table 13.* GSM8K accuracy and average context length (prompt + answer) for DeepSeek-R1-Distill-Qwen-7B. $k$: number of keys in top-$k$, $S$: SANTA sample budget. Accuracy shows 95% bootstrap confidence intervals.

| | Top-$k$ | | SANTA | |
|---|---|---|---|---|
| $k|S$ | Acc. (%) | Tok. | Acc. (%) | Tok. |
| 2 | 0.23±0.15 | 4049 | 0.00±0.00 | 4217 |
| 4 | 23.88±1.31 | 3298 | 0.08±0.09 | 4208 |
| 8 | 72.25±1.35 | 2332 | 52.46±1.57 | 2799 |
| 16 | 82.51±1.21 | 1860 | 79.00±1.28 | 1778 |
| 32 | 86.23±1.02 | 1694 | 83.19±1.23 | 1667 |
| 64 | 86.76±1.03 | 1624 | 84.99±1.09 | 1587 |
| 128 | 87.54±1.02 | 1626 | 85.42±1.00 | 1594 |
| 256 | 88.20±0.97 | 1600 | 86.93±1.01 | 1627 |
| SDPA (baseline) | | | **88.75±1.00** | 1532 |

*Table 14.* GSM8K accuracy and average context length (prompt + answer) for Llama-3.1-8B-Instruct.

| | Top-$k$ | | SANTA | | $S^2$ANTA-strat | | $S^2$ANTA-sys | |
|---|---|---|---|---|---|---|---|---|
| $k|S$ | Acc. (%) | Tok. | Acc. (%) | Tok. | Acc. (%) | Tok. | Acc. (%) | Tok. |
| 2 | 1.21±0.32 | 430 | 1.26±0.34 | 1075 | 1.01±0.32 | 1071 | 1.11±0.34 | 1071 |
| 4 | 6.27±0.72 | 583 | 1.57±0.37 | 825 | 1.54±0.38 | 611 | 1.67±0.40 | 569 |
| 8 | 41.32±1.50 | 549 | 1.44±0.38 | 207 | 1.74±0.40 | 325 | 2.10±0.45 | 340 |
| 16 | 62.88±1.57 | 430 | 5.51±0.72 | 350 | 39.12±1.50 | 352 | 44.63±1.58 | 349 |
| 32 | 72.50±1.34 | 384 | 38.26±1.43 | 348 | 67.00±1.48 | 343 | 68.59±1.42 | 339 |
| 64 | 76.12±1.32 | 358 | 63.63±1.49 | 346 | 74.43±1.31 | 343 | **76.42±1.34** | 341 |
| 128 | 77.13±1.30 | 349 | 70.23±1.42 | 341 | 75.64±1.33 | 341 | **77.33±1.34** | 342 |
| 256 | 78.19±1.33 | 340 | 75.61±1.42 | 342 | **78.17±1.28** | 343 | **77.56±1.34** | 343 |
| SDPA (baseline) | | | | | | | 78.06±1.33 | 344 |

Tables 13, 14, 15, and 16 feature additional GSM8K and MMLU results with comparisons using DeepSeek-R1-Distill-Qwen-7B and top-k attention. Table 17 exhibits long-context benchmark results identical to Table 6, except with additional top-k comparisons.

Comparing SANTA and top-k requires care, as they have dif-

*Table 15.* MMLU accuracy and average context length (prompt + answer) for DeepSeek-R1-Distill-Qwen-7B. $k$: number of keys in top-$k$, $S$: SANTA sample budget. Accuracy shows 95% bootstrap confidence intervals.

| | Top-$k$ | | SANTA | |
|---|---|---|---|---|
| $k|S$ | Acc. (%) | Tok. | Acc. (%) | Tok. |
| 2 | 24.97±1.24 | 3598 | 4.33±0.58 | 4178 |
| 4 | 38.93±1.32 | 2236 | 23.10±1.20 | 3666 |
| 8 | 55.08±1.47 | 1659 | 44.55±1.47 | 2080 |
| 16 | 59.07±1.36 | 1301 | 58.81±1.42 | 1506 |
| 32 | 60.77±1.40 | 1173 | 61.05±1.45 | 1228 |
| 64 | 62.03±1.38 | 1103 | 60.59±1.43 | 1104 |
| 128 | 61.68±1.49 | 1061 | **62.38±1.32** | 1075 |
| 256 | 62.05±1.46 | 1041 | **62.27±1.39** | 1074 |
| SDPA (baseline) | | | **63.16±1.45** | 1020 |

*Table 16.* MMLU accuracy and average context length (prompt + answer) for Llama-3.1-8B-Instruct.

| | Top-$k$ | | SANTA | | $S^2$ANTA-strat | | $S^2$ANTA-sys | |
|---|---|---|---|---|---|---|---|---|
| $k|S$ | Acc. (%) | Tok. | Acc. (%) | Tok. | Acc. (%) | Tok. | Acc. (%) | Tok. |
| 2 | 23.56±1.20 | 588 | 24.89±1.27 | 1144 | 24.28±1.21 | 1141 | 24.52±1.22 | 1141 |
| 4 | 28.37±1.27 | 359 | 25.13±1.20 | 961 | 24.65±1.26 | 807 | 24.49±1.18 | 774 |
| 8 | 39.54±1.43 | 532 | 25.06±1.23 | 284 | 25.63±1.23 | 286 | 24.47±1.23 | 292 |
| 16 | 45.26±1.37 | 484 | 26.24±1.26 | 313 | 32.46±1.32 | 350 | 34.14±1.39 | 353 |
| 32 | 49.49±1.44 | 448 | 33.81±1.34 | 354 | 43.78±1.50 | 398 | 45.48±1.45 | 403 |
| 64 | 48.33±1.37 | 433 | 41.11±1.40 | 397 | **49.25±1.48** | 415 | **48.70±1.39** | 418 |
| 128 | 49.47±1.44 | 418 | 47.46±1.45 | 412 | **49.92±1.47** | 421 | **50.60±1.37** | 421 |
| 256 | 49.75±1.43 | 421 | 49.75±1.51 | 417 | **50.90±1.49** | 422 | **51.12±1.46** | 421 |
| SDPA (baseline) | | | | | | | 49.86±1.47 | 424 |

ferent computational profiles. We present results for sample budget $S$ equal to the top-$k$ budget $k$ not as an iso-compute benchmark, but to demonstrate that SANTA achieves comparable or superior accuracy while using a fundamentally cheaper set of operations. At $S = k$: top-$k$ costs $n_q k d_k$ multiplications + $n_q k d_k$ additions, whereas SANTA costs 0 multiplications + $n_q S d_k$ additions (Table 12).

A 32-bit FP multiply costs 3.7 pJ versus 0.9 pJ for addition (Horowitz, 2014). For equal budgets $k = S$, the value-stage energy consumption for the two methods are approximately: $k \times (3.7 + 0.9) = 4.6k$ pJ per element for top-k, vs. $S \times 0.9 = 0.9S$ pJ per element for SANTA. Relative to top-k, SANTA yields an $\approx 5\times$ energy reduction for value-stage FLOPs. We acknowledge that these are approximate energy estimates; we aim to provide hardware-agnostic comparisons, as modern GPU hardware is optimized for matmuls and contiguous memory access, while SANTA involves adds and sparse memory access.

Regarding memory accesses: SANTA's accesses to elements of the $V$ matrix scale as $n_q S d_k$, which *in the worst case* matches top-k's $n_q k d_k$ memory accesses at $k = S$. However, since SANTA samples with replacement, the number of *unique* keys is $\leq S$. In practice, as we quantify in Appendix M, the number of unique keys sampled is $< S$, which

*Table 17.* Accuracy of Llama 8B on 4 long context tasks with an 8k-token prompt. Tasks: frequent word extraction (FWE), needle-in-a-haystack (NIAH), single-hop QA ($QA_1$), multi-hop QA ($QA_2$). $k$: number of keys for top-k. $S$: SANTA sample budget. Accuracy shows 95% bootstrap CIs.

| Kernel | $k\|S$ | FWE | NIAH | $QA_1$ | $QA_2$ |
|---|---|---|---|---|---|
| $S^2$ANTA-sys | 64 | 96.27±0.97 | 94.05±1.07 | 71.80±3.90 | 67.20±4.20 |
| $S^2$ANTA-sys | 128 | 97.80±0.70 | 94.15±1.02 | 68.20±4.10 | 66.80±4.00 |
| **$S^2$ANTA-sys** | **256** | **97.87±0.70** | **95.00±0.95** | **71.20±3.90** | **69.40±4.10** |
| $S^2$ANTA-strat | 64 | 95.93±0.94 | 94.35±1.15 | 71.40±4.00 | 65.40±4.20 |
| $S^2$ANTA-strat | 128 | 97.73±0.80 | 94.30±1.10 | 70.80±4.20 | 67.40±4.20 |
| **$S^2$ANTA-strat** | **256** | **98.33±0.60** | **94.55±1.05** | **71.20±4.00** | **67.00±4.20** |
| SANTA | 64 | 92.53±1.27 | 93.95±1.30 | 64.60±4.20 | 63.40±4.20 |
| SANTA | 128 | 94.53±1.13 | 91.45±1.38 | 67.80±4.10 | 63.40±4.30 |
| SANTA | 256 | 96.20±0.93 | 93.15±1.18 | 68.00±4.10 | 66.40±4.10 |
| **SDPA (baseline)** | **—** | **98.53±0.60** | **94.55±1.00** | **71.80±3.90** | **68.60±4.20** |
| top-$k$ | 64 | 93.07±1.20 | 92.55±1.10 | 65.60±4.10 | 60.80±4.10 |
| top-$k$ | 128 | 95.40±0.96 | 93.60±1.15 | 69.20±3.90 | 62.20±4.50 |
| **top-$k$** | **256** | **97.27±0.80** | **94.55±1.05** | **70.60±3.90** | **64.20±4.10** |

may offer significant caching advantages.

For benchmarking, we apply SANTA and top-k to both pre-fill and autoregressive decoding. However, in practice, these sparse attention implementations may see greater benefit during decoding, when only a single query sparsely selects rows of $V$.

SANTA variants demonstrate strong performance relative to top-k on GSM8K and MMLU benchmarks, despite SANTA requiring no multiplications and a similar number of additions and memory accesses for $k = S$. However, the most remarkable result is shown in Table 17, where $S^2$ANTA variants outperform top-k across virtually every long-context task for all $k = S$. $S^2$ANTA's lead over top-k is most apparent for multi-hop question answering ($QA_2$), where $S^2$ANTA variants lead in performance by $\approx 5\%$ in several cases. It is plausible that long context tasks — multi-hop QA in particular — have information that is not concentrated in the top-k keys, and therefore SANTA's unbiased estimate of attention outperforms top-k, which drops information and cannot attend to keys outside of the $k$ largest attention scores.

## I. Additional LongBench v2 Results

Table 18 reports the full LongBench v2 budget sweep corresponding to the compact operating-point summary in Table 2. Accuracy is reported with 95% bootstrap confidence intervals. Results are averaged over three runs; prompts longer than 32k tokens are truncated to 32k following the repository protocol. The main operating points, $S^2$ANTA-prop with $S$=128 and $S^2$ANTA-flash with $S$=2048, match SDPA within bootstrap confidence intervals in this 30k-token regime. Lower-budget $S^2$ANTA-flash settings de-

grade substantially, consistent with the sample-waste behavior discussed in Section 3.4, while $S^2$ANTA-prop remains near the SDPA baseline across the tested budgets.

*Table 18.* Full LongBench v2 sweep for Llama 8B. $S$: sample budget. Accuracy shows 95% bootstrap confidence intervals.

| Kernel | $S$ | Acc. (%) | Avg. scored tok. |
|---|---|---|---|
| $S^2$ANTA-flash | 128 | 8.748±2.964 | 30035.5 |
| $S^2$ANTA-flash | 256 | 14.049±3.504 | 30024.0 |
| $S^2$ANTA-flash | 512 | 26.574±3.740 | 29990.0 |
| $S^2$ANTA-flash | 1024 | 27.170±1.509 | 29994.2 |
| **$S^2$ANTA-flash** | **2048** | **28.827±3.422** | **29996.4** |
| $S^2$ANTA-prop | 64 | 29.092±2.534 | 29997.5 |
| **$S^2$ANTA-prop** | **128** | **28.098±1.028** | **29998.4** |
| $S^2$ANTA-prop | 256 | 28.694±1.029 | 29995.4 |
| $S^2$ANTA-prop | 1024 | 28.562±1.588 | 29996.2 |
| **SDPA (baseline)** | **—** | **28.363±1.509** | **29996.0** |

## J. Ablation Study: One-Hot Attention Samples

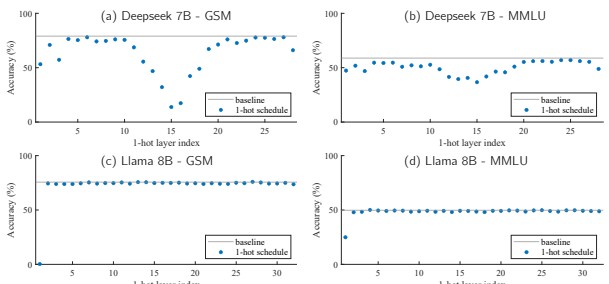

*Figure 8.* Ablations with one-hot stochastic attention. Baseline scores employ a fixed sampling budget of 16 and 256 for DeepSeek 7B and Llama 8B models, respectively. The horizontal axis shows the index of the model layer which is reduced to stochastic hard attention.

This section probes a fundamental question: how much does each attention layer matter? We find that one-hot stochastic attention, a limiting case of SANTA with $S = 1$, acts as a surprisingly sharp diagnostic. It reveals which transformer layers are robust to severe approximation and which are not, offering a practical lens into where attention precision matters most.

We perform these ablations by setting $S = 1$ in a single layer while keeping all other layers at fixed sample budgets: $S = 16$ for DeepSeek 7B (28 layers) and $S = 256$ for Llama 8B (32 layers). In this setting, each query attends to only one randomly sampled key, making that layer a hard attention bottleneck.

Models are evaluated on GSM8K and MMLU with the same protocol as Tables 4 and 5. The effect of this localized bottleneck on accuracy is summarized in Fig. 8. Some layers exhibit almost no degradation when ablated in this way, while others collapse entirely, most notably middle

layers in DeepSeek 7B and the first layer in Llama 8B.

In DeepSeek 7B, layers 12–18 are highly sensitive to approximation, while early and final layers are much more tolerant. In contrast, Llama 8B exhibits extreme sensitivity in just the first layer. These patterns suggest that attention importance is neither uniform nor trivially architectural.

We observe similar qualitative trends regardless of the baseline sample budget; the $S = 16$ and $S = 256$ settings are arbitrary. The early-layer sensitivity in Llama 8B is consistent with prior findings on high-entropy attention in shallow layers (Clark et al., 2019), and with prior work that avoids approximating early layers (Tang et al., 2024; Sun et al., 2024). The middle-layer fragility in DeepSeek 7B remains an open and intriguing observation.

## K. Layer-Wise Sample Budget Through RL

---

**Algorithm 1** Layer-wise sample-budget optimization via REINFORCE

---

**Require:** initial logits $\boldsymbol{\theta} \in \mathbb{R}^{28}$ ($\theta_i = 0$), total budget $N=224$, learning-rate $\alpha=0.02$
1: **while** training **do**
2:    **Workers (in parallel):**
3:    **for** $e = 1, \ldots, E$ **do** {$E=10$ episodes / worker run}
4:       Compute $p_i \leftarrow \exp(\theta_i)/\sum_j \exp(\theta_j)$
5:       Sample schedule $\mathbf{S} \sim \text{Multinomial}(N, \mathbf{p})$
6:       Evaluate 16 GSM8K questions using schedule $\mathbf{s}$; obtain reward $r \in [0,1]$
7:       Append $(\mathbf{s}, r)$ to shared folder
8:    **end for**
9:
10:    **Aggregator:**
11:    Collect all new $(\mathbf{s}^{(k)}, r^{(k)})$ tuples    {$k = 1 \ldots K$}
12:    Compute baseline $b \leftarrow \frac{1}{K}\sum_k r^{(k)}$
13:    **Gradient:** $\mathbf{g} \leftarrow \dfrac{1}{K}\sum_k (r^{(k)} - b)(\mathbf{s}^{(k)} - N\mathbf{p})$
14:    Update logits $\boldsymbol{\theta} \leftarrow \boldsymbol{\theta} + \alpha\,\mathbf{g}$
15:    Write back updated $\boldsymbol{\theta}$
16: **end while**

---

We now consider a global sample budget shared across all 28 transformer layers of DeepSeek 7B, allowing each layer to have a distinct sample count. We optimize this allocation using a REINFORCE-style algorithm (Williams, 1992), detailed in Algorithm 1.

We train the policy on the GSM8K training set. The policy logits $\boldsymbol{\theta} \in \mathbb{R}^{28}$ define a softmax distribution over layers. At each iteration, we sample schedules using a multinomial draw over this distribution, with a total budget of 224 samples (i.e., $8 \times 28$). Each episode evaluates 16 questions; the reward is the fraction answered correctly. We intentionally starve the baseline with $S = 8$ samples per layer to create

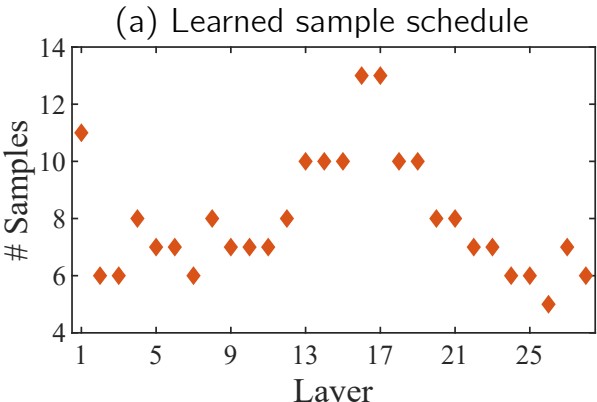

(a) Learned sample schedule

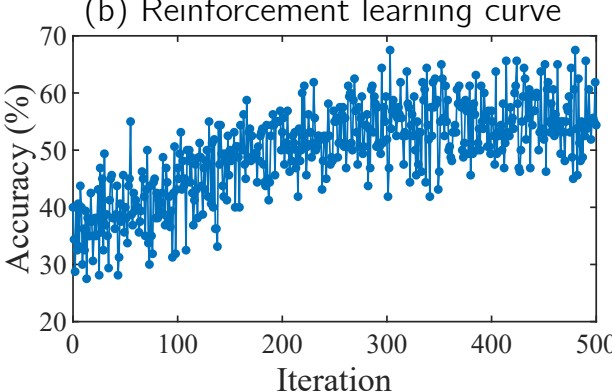

(b) Reinforcement learning curve

*Figure 9.* Learned schedule with a total budget of 224 samples across all 28 transformer blocks (DeepSeek 7B). (b) RL iterations over time.

room for the learned schedule to improve.

Fig. 9 shows the final learned schedule after 501 iterations. The allocation aligns strikingly with the ablation trends from Fig. 8: more samples are assigned to the first and middle layers, where one-hot attention was most detrimental. That REINFORCE independently rediscovers this structure from only reward signals confirms that these patterns are not artifacts of the ablation method but instead are intrinsic to the models.

This experiment is not tuned for peak accuracy, but demonstrates that adaptive per-layer budgets meaningfully outperform fixed ones. Please see Appendix L for implementation details of distributed reinforcement learning using a Kubernetes compute cluster.

*Table 19.* RL sample-schedule performance. Accuracy shows 95% bootstrap confidence intervals.

| Schedule | Task | Accuracy (%) |
|---|---|---|
| Baseline (8 samples) | GSM8K | $52.46 \pm 1.47$ |
| Baseline (8 samples) | MMLU | $44.55 \pm 1.47$ |
| Learned schedule | GSM8K | $\mathbf{66.06 \pm 1.47}$ |
| Learned schedule | MMLU | $\mathbf{48.86 \pm 1.53}$ |

## L. Distributed Reinforcement Learning on Kubernetes

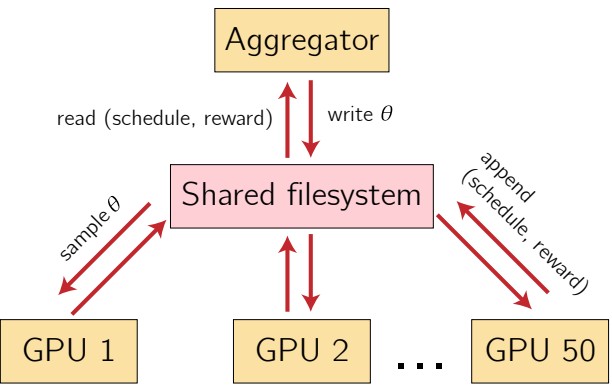

*Figure 10.* Kubernetes deployment schematic.

**Topology.** All pods mount a single read–write volume that serves as the rendezvous point.

- **Aggregator pod.** Holds the current policy and records a running history of baselines and gradients.

- **Worker pods.** Operate in a tight loop: *(i)* read the latest policy; *(ii)* sample a 28-dimensional schedule; *(iii)* evaluate ten batches of 16 questions; *(iv)* append a small JSON-lines result file to a designated "inbox" directory on the volume.

**Aggregator loop.** Whenever at least one new result file appears, the aggregator moves all current files into a private folder for the next iteration, computes the REINFORCE update and writes a new policy snapshot. It never pauses or signals the workers; it simply processes whatever has arrived. In practice, workers are unlikely to finish evaluation simultaneously, thus nearly every policy update comes from one worker output ($10 \times 16$ questions). If two land together, the aggregator just uses a double-sized batch, which lowers the gradient's variance.

**Asynchrony and fault-tolerance.**

- Workers can be terminated at any time; partial output vanishes harmlessly.

- Workers always proceed with the newest policy that has reached disk; no locking is needed.

- If the aggregator restarts it scans the history directory, resumes from the last completed iteration, and continues.

**Scaling and resources.** The worker Deployment's replica count can be scaled up or down at any time during learning. For the 501 iterations in Fig. 9 we ran 50 GPU workers on a mix of RTX 3090, L4, and A10 cards (24 GB) for roughly 20 hours, totaling $\approx 1000$ GPU-hours. The aggregator requests just one CPU core and a few hundred MiB of RAM.

**Terminology.** We keep the names aggregator and worker to highlight the simple "parameter-server" flavor of the design, but these roles align exactly with the common learner/actor split in distributed reinforcement learning.

## M. Unique Key-Count

*Table 20.* Unique-key count vs. $S$ (8192 tokens, single-query).

| $S$ | SANTA | $S^2$ANTA-strat | $S^2$ANTA-sys |
|---|---|---|---|
| 32 | 11.0 | 12.7 | 13.0 |
| 64 | 16.7 | 21.2 | 21.7 |
| 128 | 25.9 | 35.8 | 36.4 |
| 256 | 38.3 | 60.3 | 61.6 |

Since all versions of SANTA sample keys with replacement, the number of *unique* keys may be less than the sample budget $S$. In Table 20, we empirically measure the average number of unique keys sampled by the last query position for 8192-token single-hop QA prompts (normalized per head and averaged across model layers). This is indicative of the key diversity observed in autoregressive decoding. For a given $S$, $S^2$ANTA variants achieve higher key diversity than default SANTA. Furthermore, the number of unique keys tends to be $\ll S$, implying that *fewer than $S$* unique rows of $V$ require memory accesses. See Appendix N for details about the measurement protocol.

## N. Implementation & Measurement Protocol for Empirical Variance and Unique Keys Sampled

**Setting.** For a single layer/head at the last prefill position $q^\star$, let the post-softmax attention over $n_k$ keys be $p_j \geq 0$ with $\sum_{j=1}^{n_k} p_j = 1$, and let $V_j \in \mathbb{R}^{d_k}$ denote the corresponding value vectors. The dense-attention mean is

$$\mu = \sum_{j=1}^{n_k} p_j V_j \in \mathbb{R}^{d_k}.$$

**SANTA estimator.** Given a per-head sampling budget $S$, all SANTA variants estimate $\mu$ with

$$\widehat{AV} = \frac{1}{S} \sum_{s=1}^{S} V_{J_s},$$

where the key indices $\{J_s\}$ are drawn by one of three schemes: (i) multinomial; (ii) independent equal-mass stratified; or (iii) systematic (random-start, fixed-stride). These are the same variants used in the main paper; the code paths

that implement them also record the per-head statistics described below at $q^\star$.

### N.1. What We Measure (per Head at $q^\star$)

**(1) Unique keys.** The number of distinct keys touched by the sampler,

$$U := \big|\{J_1, \ldots, J_S\}\big|,$$

upper-bounded by $S$ (can be $< S$ if high-probability keys repeat). This is a proxy for how many memory locations each head actually reads.

**(2) Variance trace of the estimator.** We report the trace of the covariance (expected squared $\ell_2$ error) of $\widehat{AV}$,

$$\mathrm{VarTrace}(\widehat{AV}) := \mathbb{E}\big[\|\widehat{AV} - \mu\|_2^2\big] = \mathrm{tr}\Big(\mathrm{Cov}(\widehat{AV})\Big),$$

specialized to each sampling scheme:

- **Multinomial (closed form).** Let $\Sigma := \sum_{j=1}^{n_k} p_j\, (V_j - \mu)(V_j - \mu)^\top$ so that $\mathrm{tr}(\Sigma) = \sum_j p_j \|V_j\|_2^2 - \|\mu\|_2^2$. Then

$$\mathrm{VarTrace}_{\mathrm{multi}}(\widehat{AV}) = \frac{1}{S}\, \mathrm{tr}(\Sigma).$$

- **Independent equal-mass stratified (closed form).** Partition $[0, 1)$ into $S$ equal strata; for each stratum $m$, form the within-stratum discrete distribution induced by $p$ and let $\Sigma_m$ be the corresponding value covariance. Because draws across strata are independent,

$$\mathrm{VarTrace}_{\mathrm{strat}}(\widehat{AV}) = \frac{1}{S^2} \sum_{m=0}^{S-1} \mathrm{tr}(\Sigma_m).$$

- **Systematic (replicate-based estimate).** There is no general closed form that holds uniformly for systematic sampling, so we estimate variance by repeated random starts. With $R \geq 2$ independent offsets, we compute $\widehat{AV}^{(r)}$ for each replicate, then use

$$\widehat{\mathrm{VarTrace}}_{\mathrm{sys}}(\widehat{AV}) = \frac{1}{R-1} \sum_{r=1}^{R} \big\|\widehat{AV}^{(r)} - \overline{V}\big\|_2^2,$$

$$\overline{V} = \frac{1}{R} \sum_{r=1}^{R} \widehat{AV}^{(r)}.$$

This estimator is applied per head at $q^\star$.

### N.2. How We Aggregate and Report

For each prompt, we compute $U$ and the variance-trace scalar per head at $q^\star$, then average over heads within a layer,

average over layers, and finally average over prompts to obtain the quantities reported in the variance plots:

$$\overline{U}(S, \text{variant}) \quad \text{and} \quad \overline{T}(S, \text{variant}) = \overline{\mathrm{VarTrace}(\widehat{AV})}.$$

This yields the curves summarized in the main text as a function of the budget $S$ and the SANTA variant.

**Protocol summary.** All measurements use evaluation-mode forward passes, prefill-only, and record statistics *at the last token* per head; no additional dense attention pass is performed to form empirical errors. For multinomial and stratified we use the exact formulas above; for systematic we use the replicate-based estimator.

## O. Prompting, Parsing, and Grading

In Section 4.1, we evaluate the GSM8K test split (1319 prompts) 3 times. Evaluation code uses straightforward answer parsing and borrows grading code from PRM800K (Lightman et al., 2024). For all runs, temperature $= 0.6$, top-p $= 0.95$, and repetition penalty $= 1.1$. We use left-padded batches of 8–16 prompts.

In Section 4.2, we average results over three runs on the MMLU validation set (1531 questions), using identical decoding parameters as GSM8K.

In Section 4.3, long-context benchmark results are averaged over 500 prompts and greedy decoding is employed with no repetition penalty applied.

In Section 3.4 and Appendix I, LongBench v2 (Bai et al., 2025) results are averaged over three runs. Prompts longer than 32k tokens are truncated to 32k following the repository protocol. We report accuracy with 95% bootstrap confidence intervals and the average number of scored tokens.

### DeepSeek 7B + GSM8K

| Aspect | Procedure |
|---|---|
| Prompt | Question, blank line, then `Please reason step by step, and put your final numeric answer within \boxed{}.` |
| Answer extraction | Perform a brace-balanced search for the last occurrence of `\boxed{...}` (robust to nested braces). If none is found, fall back to the final $\sim$200 characters of the model output. |
| Grading | Numeric equality judged by the MIT-licensed PRM800K grader; every prompt is counted, and blank or unparsable predictions score 0. |

*Table 21.* Prompting and grading for DeepSeek 7B on GSM8K.

### DeepSeek 7B + MMLU

| Aspect | Procedure |
|---|---|
| Prompt | Question, four labeled choices, then `Answer briefly.  Put your final answer as a single letter inside \boxed{}.` |
| Answer extraction | Brace-balanced search for the last occurrence of `\boxed{...}`; if none is found, inspect the final ∼300 characters of the output. Scan this region backwards and take the first occurrence of A–D (case-insensitive), then convert it to uppercase. |
| Grading | The predicted letter is compared with the gold key (case-insensitive). If no valid letter is found—or the output is blank—the item scores 0; otherwise it scores 1 when the letters match. |

*Table 22.* Prompting and grading for DeepSeek 7B on MMLU.

### Llama 8B + GSM8K

| Aspect | Procedure |
|---|---|
| Prompt | Meta chat template. *System*: "You are a precise mathematician. Explain your reasoning step by step, then output ONLY the final number on a new line." *User*: the GSM8K question. |
| Answer extraction | Inspect the model reply in order: (i) the last non-empty line of the response; (ii) if that fails, the final 25 whitespace-separated tokens. From the selected chunk, take the first integer-looking token (commas allowed, optional leading minus). |
| Grading | Numeric equality judged by the MIT-licensed PRM800K grader; every question is counted, and blank or unparsable answers score 0. |

*Table 23.* Prompting and grading for Llama 8B on GSM8K.

### Llama 8B + MMLU

| Aspect | Procedure |
|---|---|
| Prompt | Meta chat template. *System*: "You are a knowledgeable and concise subject-matter expert. Work through the problem step by step. Finally, on a new line, output ONLY the single capital letter (A, B, C, or D) that corresponds to the correct choice." *User*: the question followed by the four labeled choices A–D. |
| Answer extraction | Take the substring after the last newline and scan it left-to-right for the first capital A–D (case-insensitive). If none is found, inspect the final 10 whitespace-separated tokens of the whole response, scanning them right-to-left for A–D. |
| Grading | Predicted letter (upper-cased) is compared with the gold key. Every question is counted; if no valid letter is found or it does not match the gold answer, the item scores 0. |

*Table 24.* Prompting and grading for Llama 8B on MMLU.

### Llama 8B + RULER prompts

| Aspect | Procedure |
|---|---|
| Prompt | Prompts are generated for the following RULER sub-tasks with a length of 4096–32768 tokens: fwe, niah_multivalue, qa_1, and qa_2. Greedy decoding is used. The model generates a maximum of 64 new tokens. |
| Grading | fwe: partial credit = (# of gold words present in the model output) / 3 (case-insensitive set match). niah_multivalue: take the first four unique integers from the model output; partial credit = (# of these that appear in the gold set) / 4. qa_1 and qa_2: correct if the gold answer string appears as a case-insensitive substring of the model output. |

*Table 25.* Prompting and grading for Llama 8B on RULER prompts.

**Licenses.**

- **Models.** deepseek-ai/DeepSeek-R1-Distill-Qwen-7B (MIT) and meta-llama/Meta-Llama-3.1-8B-Instruct (Llama 3.1 license) on Hugging Face.
- **Datasets.** GSM8K openai/gsm8k (MIT) and MMLU cais/mmlu (MIT) on Hugging Face. RULER prompts (Apache).
- **Grader.** PRM800K numeric grader (MIT).

## P. $S^2$ANTA-Prop Pseudocode

Algorithm 2 provides a high-level overview of $S^2$ANTA-prop. Algorithms 3, 4, and 5 show breakdowns of the kernels that perform score computation, sample budget allocation,

and the $V$ gather, respectively.

---

**Algorithm 2** S$^2$ANTA-prop (overview, decode/GQA): proportional tile budgets + systematic sparse-$V$

---

**Require:** $Q \in \mathbb{R}^{H \times d_k}$; $K, V \in \mathbb{R}^{n_k \times H_{kv} \times d_k}$ (GQA); tile length $B_{tile}$; samples/head $S$.
**Ensure:** $O \in \mathbb{R}^{H \times d_k}$ approximating softmax$(QK^\top)V$.
1: $T \leftarrow \lceil n_k/B_{tile} \rceil$, $G \leftarrow H/H_{kv}$, $k(h) \leftarrow \lfloor h/G \rfloor$, $\mathcal{T}_t = [tB_{tile}, \min((t+1)B_{tile}, n_k))$.
2: **Pass 1:** compute $m_{h,t} = \max_{n \in \mathcal{T}_t} s_{h,n}$ and $\ell_{h,t} = \sum_{n \in \mathcal{T}_t} \exp(s_{h,n} - m_{h,t})$; optionally stash $u_{h,n} = \exp(s_{h,n} - m_{h,t})$, where $s_{h,n} = Q_h^\top K_{n,k(h)}/\sqrt{d_k}$.
3: **Budgets:** $m_h^\star = \max_t m_{h,t}$; $W_{h,t} = \exp(m_{h,t} - m_h^\star)\ell_{h,t}$; allocate integers $\{S_{h,t}\}_t$ with $\sum_t S_{h,t} = S$ and $S_{h,t} \approx S \cdot W_{h,t}/\sum_t W_{h,t}$ (largest remainder); set $\text{inv}\delta_{h,t} = S_{h,t}/\ell_{h,t}$ and $a_{0,h,t} \sim U[0,1)$.
4: **Pass 2:** systematic counts $c_{h,n}$ from $x_{h,n} = \text{inv}\delta_{h,t}u_{h,n}$, then $O_h \leftarrow O_h + \sum_n c_{h,n} V_{n,k(h)}$.
5: **return** $O/S$.

---

**Algorithm 3** S$^2$ANTA-prop, Kernel 1: pass-1 tile statistics (and optional $u$-stash)

---

**Require:** $Q \in \mathbb{R}^{H \times d_k}$; $K \in \mathbb{R}^{n_k \times H_{kv} \times d_k}$; tile length $B_{tile}$.
**Ensure:** $m \in \mathbb{R}^{H \times T}$, $\ell \in \mathbb{R}^{H \times T}$, and optional $u \in \mathbb{R}^{H \times n_k}$.
1: $T \leftarrow \lceil n_k/B_{tile} \rceil$; $G \leftarrow H/H_{kv}$.
2: **for** each KV head $k \in \{0, \dots, H_{kv}-1\}$ and tile $t \in \{0, \dots, T-1\}$ **in parallel do**
3: $\quad \mathcal{T}_t \leftarrow [tB_{tile}, \min((t+1)B_{tile}, n_k))$.
4: $\quad$ **for** each grouped head $g \in \{0, \dots, G-1\}$ **do**
5: $\quad\quad h \leftarrow kG + g$; $s_{h,n} \leftarrow Q_h^\top K_{n,k}/\sqrt{d_k}$ for all $n \in \mathcal{T}_t$.
6: $\quad\quad m_{h,t} \leftarrow \max_{n \in \mathcal{T}_t} s_{h,n}$; $\ell_{h,t} \leftarrow \sum_{n \in \mathcal{T}_t} \exp(s_{h,n} - m_{h,t})$.
7: $\quad\quad$ (Optional) $u_{h,n} \leftarrow \exp(s_{h,n} - m_{h,t})$ for all $n \in \mathcal{T}_t$.
8: $\quad$ **end for**
9: **end for**

---

**Algorithm 4** S$^2$ANTA-prop, Kernel 2: proportional per-tile budgets (largest remainder)

---

**Require:** Tile stats $m_{h,t}, \ell_{h,t}$; samples/head $S$; seed.
**Ensure:** Integer budgets $S_{h,t}$ with $\sum_t S_{h,t} = S$, plus $\text{inv}\delta_{h,t}$ and offsets $a_{0,h,t}$.
1: **for** each head $h \in \{0, \dots, H-1\}$ **in parallel do**
2: $\quad m_h^\star \leftarrow \max_t m_{h,t}$.
3: $\quad W_{h,t} \leftarrow \exp(m_{h,t} - m_h^\star)\ell_{h,t}$; $Z_h \leftarrow \sum_{t=0}^{T-1} W_{h,t}$.
4: $\quad q_t \leftarrow S \cdot W_{h,t}/Z_h$; $S_{h,t} \leftarrow \lfloor q_t \rfloor$ for all $t$.
5: $\quad$ Give remaining $S - \sum_t S_{h,t}$ samples to tiles with largest fractional part of $q_t$ (largest remainder).
6: $\quad \text{inv}\delta_{h,t} \leftarrow (S_{h,t} > 0)\ ?\ S_{h,t}/\ell_{h,t}\ :\ 0$; $a_{0,h,t} \sim \text{Unif}(0,1)$.
7: **end for**

---

**Algorithm 5** S$^2$ANTA-prop, Kernel 3: systematic resampling and sparse-$V$ accumulation (GQA grouped)

---

**Require:** $V \in \mathbb{R}^{n_k \times H_{kv} \times d_k}$; $u_{h,n}$; $\text{inv}\delta_{h,t}, a_{0,h,t}$; tile length $B_{tile}$.
**Ensure:** Accumulator $O \in \mathbb{R}^{H \times d_k}$ (fp32).
1: $G \leftarrow H/H_{kv}$; $O \leftarrow 0$.
2: **for** each KV head $k \in \{0, \dots, H_{kv}-1\}$ and tile $t \in \{0, \dots, T-1\}$ **in parallel do**
3: $\quad \mathcal{T}_t \leftarrow [tB_{tile}, \min((t+1)B_{tile}, n_k))$; heads $\mathcal{H}_k = \{kG, \dots, (k+1)G-1\}$.
4: $\quad$ **for** each head $h \in \mathcal{H}_k$ **do**
5: $\quad\quad p \leftarrow 0$.
6: $\quad\quad$ **for** each $n \in \mathcal{T}_t$ (increasing) **do**
7: $\quad\quad\quad x \leftarrow \text{inv}\delta_{h,t}\, u_{h,n}$.
8: $\quad\quad\quad c_{h,n} \leftarrow \lfloor a_{0,h,t} + p + x \rfloor - \lfloor a_{0,h,t} + p \rfloor$; $p \leftarrow p + x$.
9: $\quad\quad$ **end for**
10: $\quad$ **end for**
11: $\quad$ (Optional) compact emitted rows $\mathcal{E}_t = \{n \in \mathcal{T}_t : \exists h \in \mathcal{H}_k, c_{h,n} > 0\}$.
12: $\quad$ **for** each $n \in \mathcal{E}_t$ and each $h \in \mathcal{H}_k$ **do**
13: $\quad\quad O_h \leftarrow O_h + c_{h,n} V_{n,k}$.
14: $\quad$ **end for**
15: **end for**
16: **return** $O$ (apply final scaling by $1/S$ outside or in a final cast/scale kernel).

---

## Q. $S^2$**ANTA-Flash Pseudocode**

Algorithm 6 provides a high-level overview of $S^2$ANTA-flash. Algorithms 7 and 8 describe the 2 kernels that constitute $S^2$ANTA-flash in detail.

---

**Algorithm 6** $S^2$**ANTA-flash** (overview, decode/GQA): uniform per-tile sampling + FlashDecoding-style merge

---

**Require:** $Q \in \mathbb{R}^{H \times d_k}$; $K, V \in \mathbb{R}^{n_k \times H_{\text{kv}} \times d_k}$ (GQA); tile length $B_{\text{tile}}$; samples/head $S$; seed.

**Ensure:** $O \in \mathbb{R}^{H \times d_k}$ approximating $\text{softmax}(QK^\top)V$.

1: $T \leftarrow \lceil n_k/B_{\text{tile}} \rceil$; $G \leftarrow H/H_{\text{kv}}$; $k(h) \leftarrow \lfloor h/G \rfloor$; uniform per-tile budget $S_{\text{tile}} \approx S/T$ (rounded).

2: **Kernel 1 (tile partials):** for each tile $(k, t)$, compute $(m_{h,t}, \ell_{h,t})$ and $\widetilde{O}_{h,t} = \sum_{n \in \mathcal{T}_t} c_{h,n} V_{n,k}$ via systematic sampling with budget $S_{\text{tile}}$.

3: **Kernel 2 (merge):** for each head $h$, set $m_h^\star = \max_t m_{h,t}$, $W_{h,t} = \exp(m_{h,t} - m_h^\star) \ell_{h,t}$, $Z_h = \sum_t W_{h,t}$, and $O_h = \frac{1}{Z_h} \sum_t W_{h,t} \left( \widetilde{O}_{h,t}/S_{\text{tile}} \right)$.

4: **return** $O$.

---

**Algorithm 7** $S^2$ANTA-flash, Kernel 1: fused per-tile partials (stats + systematic sampling)

---

**Require:** $Q \in \mathbb{R}^{H \times d_k}$; $K, V \in \mathbb{R}^{n_k \times H_{\text{kv}} \times d_k}$; tile length $B_{\text{tile}}$; samples/head $S$; seed.

**Ensure:** Per-tile outputs $(m_{h,t}, \ell_{h,t}, \widetilde{O}_{h,t})$ for all heads and tiles.

1: $T \leftarrow \lceil n_k/B_{\text{tile}} \rceil$; $G \leftarrow H/H_{\text{kv}}$; $S_{\text{tile}} \approx S/T$ (rounded).

2: **for** each KV head $k \in \{0, \dots, H_{\text{kv}}-1\}$ and tile $t \in \{0, \dots, T-1\}$ **in parallel do**

3: $\quad \mathcal{T}_t \leftarrow [tB_{\text{tile}}, \min((t+1)B_{\text{tile}}, n_k))$; heads $\mathcal{H}_k = \{kG, \dots, (k+1)G-1\}$.

4: $\quad$ **for** each head $h \in \mathcal{H}_k$ **do**

5: $\qquad s_{h,n} \leftarrow Q_h^\top K_{n,k}/\sqrt{d_k}$ for $n \in \mathcal{T}_t$.

6: $\qquad m_{h,t} \leftarrow \max_{n \in \mathcal{T}_t} s_{h,n}$; $\ u_{h,n} \leftarrow \exp(s_{h,n} - m_{h,t})$; $\ell_{h,t} \leftarrow \sum_{n \in \mathcal{T}_t} u_{h,n}$.

7: $\qquad \text{inv}\delta_{h,t} \leftarrow (S_{\text{tile}}/\ell_{h,t})$ (or 0 if $\ell_{h,t}$ is tiny); $\ a_{0,h,t} \sim U[0, 1)$.

8: $\qquad$ Use systematic counts $c_{h,n}$ from $x_{h,n} = \text{inv}\delta_{h,t} u_{h,n}$ and accumulate $\widetilde{O}_{h,t} \leftarrow \sum_{n \in \mathcal{T}_t} c_{h,n} V_{n,k}$.

9: $\quad$ **end for**

10: **end for**

---

**Algorithm 8** $S^2$ANTA-flash, Kernel 2: FlashAttention-style merge across tiles

---

**Require:** Per-tile $(m_{h,t}, \ell_{h,t}, \widetilde{O}_{h,t})$ from Alg. 7; $S_{\text{tile}}$.

**Ensure:** Final $O \in \mathbb{R}^{H \times d_k}$.

1: **for** each head $h \in \{0, \dots, H-1\}$ **in parallel do**

2: $\quad m_h^\star \leftarrow \max_t m_{h,t}$.

3: $\quad W_{h,t} \leftarrow \exp(m_{h,t} - m_h^\star) \ell_{h,t}$; $\ Z_h \leftarrow \sum_{t=0}^{T-1} W_{h,t}$.

4: $\quad O_h \leftarrow \frac{1}{Z_h} \sum_{t=0}^{T-1} W_{h,t} \left( \widetilde{O}_{h,t}/S_{\text{tile}} \right)$.

5: **end for**

6: **return** $O$.

# R. Decode-Step Kernel Microbenchmark Details

This appendix describes the microbenchmark used to measure *decode-step* attention latency for FlashAttention2 decoding ("FlashDecoding"), FlashInfer decoding, and our custom $S^2$ANTA CUDA kernels. The intent is to be explicit about tensor shapes, software/hardware environment, cache state, and what is (and is not) included in the reported runtimes, in order to support a fair comparison.

**What is being measured.** We measure GPU kernel time for a single decoding step (one new token) as a function of the KV-cache length $n_k$. Timing is performed with CUDA events, recorded on the active CUDA stream and synchronized on the end event each iteration. The reported milliseconds therefore reflect elapsed time on the GPU timeline for the CUDA work launched by a single backend call (which may internally launch multiple kernels). All Python/CPU overhead and input construction are excluded by allocating and populating tensors outside the timed region and recording events immediately around the backend call.

**Hardware and driver.** All measurements were run on a single **NVIDIA RTX 6000 Ada Generation** GPU (48 GB). The installed driver reported by `nvidia-smi` was 580.82.07 (CUDA compatibility reported as 13.0).

**Software environment (versions).** The benchmark environment used:

- **PyTorch 2.6.0+cu124**

- **flash-attn 2.7.1** (FlashAttention2)

- **flashinfer-python 0.5.3**

Our $S^2$ANTA kernels are invoked through a PyTorch CUDA extension (custom CUDA code callable from PyTorch); the timed region includes all CUDA kernels launched by this extension entrypoint.

**Decode geometry and tensor shapes.** We benchmark a single-batch decode step with head geometry matching a Llama-style GQA configuration:

$$H = 32, \quad H_{kv} = 8, \quad d_k = 128, \quad G = H/H_{kv} = 4,$$

where $H$ is the number of query heads, $H_{kv}$ is the number of KV heads, and $d_k$ is the head dimension. The query length is one token (decode), batch size is one, and attention is causal.

For each cache length $n_k$, the following tensors are constructed once and reused across timing iterations:

- Query tensor with shape $[H, d_k]$.

- Key/value cache tensors stored in a sequence-major layout with shape $[n_k+1, H_{kv}, d_k]$. (The extra $+1$ slot corresponds to the current token.)

All backends are run with the same causal setting and the standard softmax scaling $1/\sqrt{d_k}$, with dropout disabled.

**Input distribution and dtype.** All inputs are i.i.d. Gaussian (`torch.randn`) generated directly on the GPU. Unless otherwise stated, the query, keys, and values are provided in bfloat16 (the harness also supports fp16). A fixed RNG seed is used so that, for a given $n_k$, each backend sees the same input tensors.

**Cold-cache protocol (L2 thrashing).** To reduce sensitivity to favorable cache residency effects: before *each* timed iteration, we touch a large GPU buffer by performing an in-place update on a 512 MB float32 tensor. This operation is intended to thrash GPU caches (including L2) via a large read+write stream. The cache-thrash operation is executed *outside* the timed region (before the start event is recorded), and therefore is not included in the reported kernel time.

**Timing protocol and trial structure.** For each cache length $n_k$ and each backend, we run a fixed number of warmup and timed iterations:

- **Warmup:** 10 untimed iterations to amortize one-time effects (e.g., kernel selection/caching inside libraries).

- **Timing:** 40 timed iterations. Each iteration performs an optional cache-thrash (not timed), then records a start CUDA event, launches the backend once, records an end CUDA event, synchronizes on the end event, and reads elapsed milliseconds from the event pair.

- **Aggregation:** we report the mean latency over the 40 timed iterations.

All tensor allocations and concatenations/appends used to construct inputs are performed outside the timed region.

**Note on KV-cache appends in FlashDecoding.** FlashAttention2 decoding performs a fused KV-cache append/update as part of its decode call, whereas FlashInfer decoding and our $S^2$ANTA decode kernels do not perform an internal append in this benchmark. To isolate the attention-like decode computation for FlashInfer and $S^2$ANTA, we provide a cache tensor that already includes the current token in the preallocated $[n_k+1, H_{kv}, d_k]$ storage.

This asymmetry is not material in the long-context regime: the fused append cost is $O(H_{kv}d_k)$ per step and does not

scale with $n_k$, so its fractional contribution decreases as $n_k$ grows. Our speedup comparisons focus on large KV-cache lengths (e.g., 8k–32k tokens), where the attention computation dominates. Including the fused append inside FlashDecoding therefore does not change qualitative conclusions in this regime (and makes the FlashDecoding measurement slightly more conservative relative to an "attention-only" timing).

## S. Additional $S^2$ANTA Kernel Results

*Table 26.* Accuracy of Llama 8B on 4 long-context tasks with an 8k-token prompt. Tasks: frequent word extraction (FWE), needle-in-a-haystack (NIAH), single-hop QA ($QA_1$), multi-hop QA ($QA_2$). $S$: sample budget. Accuracy shows 95% bootstrap confidence intervals.

| Kernel | $S$ | FWE | NIAH | $QA_1$ | $QA_2$ |
|---|---|---|---|---|---|
| $S^2$ANTA-prop | 64 | 96.60±0.52 | 93.23±0.69 | 70.73±2.43 | 67.33±2.46 |
| **$S^2$ANTA-prop** | **128** | **98.04±0.40** | **93.78±0.60** | **70.33±2.40** | **66.87±2.43** |
| $S^2$ANTA-prop | 256 | 98.07±0.40 | 94.05±0.62 | 70.67±2.33 | 66.40±2.30 |
| $S^2$ANTA-flash | 64 | 66.47±1.12 | 83.57±1.05 | 65.67±2.44 | 62.27±2.40 |
| $S^2$ANTA-flash | 128 | 86.11±0.91 | 91.70±0.73 | 69.47±2.30 | 66.27±2.50 |
| $S^2$ANTA-flash | 256 | 94.80±0.62 | 92.77±0.63 | 70.67±2.20 | 66.80±2.50 |
| **$S^2$ANTA-flash** | **1024** | **98.11±0.40** | **94.22±0.58** | **70.73±2.30** | **66.33±2.40** |
| **SDPA (baseline)** | — | **98.53±0.60** | **94.55±1.00** | **71.80±3.90** | **68.60±4.20** |

## T. HumanEval Code-Generation Results

Table 27 reports HumanEval (Chen et al., 2021) pass@1 accuracy for Llama-3.1-8B-Instruct using the $S^2$ANTA-prop and $S^2$ANTA-flash kernels. We include HumanEval as an additional benchmark-diversity check. Because HumanEval prompts are short relative to the long-context decode regime targeted by SANTA, these results are intended to characterize accuracy rather than demonstrate kernel speedups. At $S = 256$, both $S^2$ANTA-flash and $S^2$ANTA-prop remain within the bootstrap confidence interval of the SDPA baseline.

*Table 27.* HumanEval pass@1 accuracy for Llama 8B. $S$: sample budget. Pass@1 shows 95% bootstrap confidence intervals.

| Method | $S$ | Pass@1 (%) | Avg. tok. |
|---|---|---|---|
| $S^2$ANTA-flash | 16 | 12.805±4.370 | 521.6 |
| $S^2$ANTA-flash | 32 | 44.106±6.707 | 296.5 |
| $S^2$ANTA-flash | 64 | 59.350±6.504 | 267.7 |
| $S^2$ANTA-flash | 128 | 60.569±6.809 | 263.0 |
| **$S^2$ANTA-flash** | **256** | **63.008±6.911** | **260.1** |
| $S^2$ANTA-prop | 16 | 46.545±6.301 | 274.6 |
| $S^2$ANTA-prop | 32 | 61.789±6.504 | 267.1 |
| $S^2$ANTA-prop | 64 | 60.366±7.012 | 262.5 |
| $S^2$ANTA-prop | 128 | 62.398±6.911 | 262.5 |
| **$S^2$ANTA-prop** | **256** | **63.211±7.012** | **261.1** |
| **SDPA (baseline)** | — | **65.244±7.317** | **261.4** |

**Protocol.** We evaluate the standard HumanEval pass@1 setting with greedy decoding and compare model-generated completions against the unit tests for each problem. We report pass@1 with 95% bootstrap confidence intervals and the average number of generated/evaluated tokens. Since HumanEval examples are short-context code-generation tasks, they do not exercise the memory-bound 30k–32k-token decode regime emphasized in Section 3.4. Accordingly, we use HumanEval only to verify that the approximation does not catastrophically degrade code-generation accuracy at moderate budgets.

## U. Same-State Fidelity and Tile-Sensitivity Analyses

This appendix reports additional analyses to characterize when the $S^2$ANTA kernels are most faithful to dense attention. We focus on 32k-token frequent-word extraction (FWE) and multi-hop QA ($QA_2$) prompts, since these are representative long-context tasks used in the main kernel-accuracy evaluation. Unless otherwise stated, we select 40 long prompts per task and examine 32 decoding steps per prompt. At each step, using the dense model's current prefix as the reference context, we apply the $S^2$ANTA approximation only in the final decoding layer and compare the resulting attention output $AV$ against the corresponding dense-attention output at the same decode state. Relative L2 error and cosine similarity are averaged over attention heads and evaluated decoding steps. Same-state top-1 agreement is measured by running a full one-token forward pass and checking whether the approximate model preserves the dense model's top-1 next-token prediction.

### U.1. Numerical Approximation Error

Table 28 reports same-state fidelity for $S^2$ANTA-flash and $S^2$ANTA-prop. Approximation quality improves with the sample budget $S$. The low same-state top-1 agreement for $S^2$ANTA-flash at $S = 256$ is consistent with its downstream accuracy drop in Table 1. By contrast, the main operating points used in Section 3.4, namely $S = 2048$ for $S^2$ANTA-flash and $S = 128$ for $S^2$ANTA-prop, recover high cosine similarity and high top-1 next-token agreement.

### U.2. Sensitivity of $S^2$ANTA-Flash to Attention Support Across Tiles

Table 29 studies how $S^2$ANTA-flash fidelity changes as attention mass spreads across more tiles. We fix the $S^2$ANTA-flash sample budget to $S = 2048$, matching the main 32k-token operating point in Section 3.4, and use tile size 256. At each decode position, we compute the dense attention distribution and count how many tiles are needed to capture 90% of total attention mass. We then bucket observations

*Table 28.* Same-state numerical fidelity of $S^2$ANTA-flash and $S^2$ANTA-prop on 32k-token FWE and QA$_2$ prompts. Relative L2 error and cosine similarity compare final-layer sparse and dense attention outputs $AV$ at matched decode states. Top-1 agreement reports whether a full one-token approximate forward pass preserves the dense model's top-1 next-token prediction.

| Task | Backend | $S$ | Mean relative L2 error | Mean cosine similarity | Same-state top-1 agreement |
|---|---|---|---|---|---|
| FWE | flash | 256 | 0.244 | 0.963 | 91% |
| FWE | flash | 1024 | 0.091 | 0.993 | 97% |
| FWE | flash | 2048 | 0.055 | 0.997 | 97% |
| FWE | prop | 128 | 0.131 | 0.986 | 96% |
| FWE | prop | 256 | 0.085 | 0.994 | 97% |
| FWE | prop | 1024 | 0.035 | 0.999 | 98% |
| QA$_2$ | flash | 256 | 0.572 | 0.851 | 91% |
| QA$_2$ | flash | 1024 | 0.233 | 0.968 | 97% |
| QA$_2$ | flash | 2048 | 0.144 | 0.986 | 99% |
| QA$_2$ | prop | 128 | 0.208 | 0.967 | 99% |
| QA$_2$ | prop | 256 | 0.141 | 0.983 | 97% |
| QA$_2$ | prop | 1024 | 0.063 | 0.996 | 99% |

by this tile-support statistic.

$S^2$ANTA-flash is most accurate when most attention mass is concentrated in a small number of tiles. Fidelity degrades as attention mass becomes more diffuse, especially on QA$_2$, where relevant evidence may be spread across the context. This behavior is consistent with the "sample waste" explanation in Section 3.4: uniform per-tile sampling spends samples on low-mass tiles before deferred renormalization.

*Table 29.* Sensitivity of $S^2$ANTA-flash to attention support spread across tiles. Tile size is fixed to 256 and sample budget is fixed to $S = 2048$. "Tiles to reach 90% attention mass" is computed from the dense attention distribution at each evaluated decode state.

| Task | Tiles to reach 90% attention mass | Mean relative L2 error | Mean cosine similarity |
|---|---|---|---|
| FWE | 1 tile | 0.035 | 0.998 |
| FWE | 2 tiles | 0.047 | 0.998 |
| FWE | 3–4 tiles | 0.068 | 0.997 |
| FWE | 5–8 tiles | 0.084 | 0.995 |
| FWE | 9+ tiles | 0.073 | 0.996 |
| QA$_2$ | 1 tile | 0.055 | 0.997 |
| QA$_2$ | 2 tiles | 0.117 | 0.991 |
| QA$_2$ | 3–4 tiles | 0.170 | 0.982 |
| QA$_2$ | 5–8 tiles | 0.175 | 0.981 |
| QA$_2$ | 9+ tiles | 0.190 | 0.978 |

## U.3. Sensitivity of $S^2$ANTA-Flash to Tile Size

Table 30 reports a tile-size sweep using the same same-state fidelity protocol. We fix the sequence length to 32k tokens and the $S^2$ANTA-flash sample budget to $S = 2048$. Smaller tile sizes create more total tiles and therefore increase the amount of uniform per-tile sampling spent on low-mass regions of the attention distribution. Larger tiles are substantially more faithful under this fixed-budget setting, especially on QA$_2$.

*Table 30.* Sensitivity of $S^2$ANTA-flash to tile size at 32k-token contexts. Sample budget is fixed to $S = 2048$. Smaller tiles imply more total tiles and more opportunity for sample waste under uniform per-tile sampling.

| Task | Tile size | Mean relative L2 error | Mean cosine similarity |
|---|---|---|---|
| FWE | 32 | 0.145 | 0.985 |
| FWE | 64 | 0.119 | 0.989 |
| FWE | 256 | 0.055 | 0.997 |
| QA$_2$ | 32 | 0.390 | 0.918 |
| QA$_2$ | 64 | 0.269 | 0.958 |
| QA$_2$ | 256 | 0.144 | 0.986 |

Together, Tables 28–30 show that fidelity improves with sample budget, that the main $S^2$ANTA operating points recover high same-state agreement, and that $S^2$ANTA-flash is most faithful when attention mass is concentrated in relatively few tiles. These observations suggest that dynamic tile-aware sample allocation could further improve the accuracy–latency trade-off beyond the static budgets studied in this work.

