# OpenReview forum: "Stochastic Sparse Attention for Memory-Bound Inference"
_ICML.cc/2026/Conference — ICML 2026 regular_

### Official Review · Reviewer_Tb8z · 2026-03-10

**Soundness:** 3
**Presentation:** 3
**Significance:** 3
**Originality:** 3
**Overall Recommendation:** 5
**Confidence:** 3

**Summary:**

Autoregressive decoding at long context lengths is bottlenecked by the memory bandwidth required to stream the full KV cache for every generated token. This paper introduces Stochastic Additive No-mulT Attention (SANTA), which sparsifies value-cache access by sampling S indices from the post-softmax attention distribution and aggregating only those value rows via gather-and-add. The paper proposes variance-reduced variants (S2ANTA), and additionally introduces Bernoulli qKT sampling as a complementary technique that sparsifies the score stage by reducing key-feature access during decoding. Since the sampling step introduces a sequential dependency that prevents straightforward FlashDecoding-style parallelization, the paper proposes two GPU kernel strategies — S2ANTA-prop and S2ANTA-flash — to enable efficient parallel execution.

**Compliance With Llm Reviewing Policy:**

Affirmed.

**Final Justification:**

During the rebuttal, the authors satisfactorily addressed all of my concerns. They provided evaluations and reported end-to-end speedups. Their discussion of the impact of KV-cache quantization was also convincing, and they raised an interesting point that quantization proportionally compresses model weights as well and, in some cases, may actually favor their method.

**Key Questions For Authors:**

(1) Which sampling method do the evaluations in Figure 4 and Table 1 use - S2ANTA-strat or S2ANTA-sys?

(2) What datatype is used for the KV cache in the kernel benchmarks (Figure 4)?

(3) The paper argues that SANTA is orthogonal to KV-cache quantization. I'm curious how the two interact in practice: when KV caches are already quantized to low bitwidths, the per-element memory cost is much lower, so does the absolute bandwidth saving from sparse access shrink proportionally? Conversely, there may be an accuracy advantage to keeping the KV cache in higher precision and relying on SANTA's sparsification for bandwidth reduction instead of quantizing. It would be interesting to see these tradeoffs.

(4) Given that the reported 1.5x speedup is at the attention-kernel level, what fraction of end-to-end decode latency does the attention kernel typically account for in the 32k-token regime, and how much of the 1.5x kernel speedup do you expect to survive once MLP, LayerNorm, and other components are included? Also, as GQA group sizes grow beyond 4, each unique KV head serves more query heads — does this reduce the effective benefit of sparse V access?

(5) The kernel benchmarks use an RTX 6000 Ada. Since SANTA's speedup is driven by reducing memory traffic, how do you expect the results to translate to datacenter GPUs like the A100 or H100, where the bandwidth-to-compute ratio is higher?

**Limitations:**

yes

**Strengths And Weaknesses:**

Strengths:
- Unlike cache eviction or token pruning methods, SANTA reduces the amount of KV data accessed per decode step without permanently discarding cache contents, preserving the ability to attend to any token.
- The paper is well written and easy to follow.
- The paper addresses the systems challenge that sampling creates a sequential dependency incompatible with split-KV parallelization, and proposes two concrete kernel strategies (prop and flash) to overcome it.
- Thorough empirical coverage across reasoning (GSM8K, MMLU) and long-context (RULER) tasks.

Weaknesses:
- Only kernel-level speedups are reported (1.5x on the attention kernel); no end-to-end inference latency or throughput measurements are provided. The paper acknowledges Amdahl's Law applies, but without end-to-end numbers it is difficult to assess the practical impact.
- The method's memory-access benefits are limited to the decode phase, so the practical impact depends on having long contexts and decode-heavy workloads.
- Kernel benchmarks are conducted solely on an RTX 6000 Ada, a workstation GPU. Datacenter inference GPUs (A100, H100) have significantly different memory bandwidth and compute profiles, so the reported 1.5x speedup may not transfer directly to the hardware where long-context LLM serving is most commonly deployed.

---

> ### Author Rebuttal · Authors · 2026-03-31
>
> We thank the reviewer for the thoughtful feedback and engagement.
>
> ### Response to Weakness 1: On End-to-End Speedup
>
> We integrated our kernels with HuggingFace generate(), showing end-to-end speedups for batch inference. We consider representative QA prompts, truncated to 30k tokens for batch inference, for a batch size of 6 on RTX 6000 Ada, decoding 2k new tokens. Sample budgets $S$ match our headline results which show $\approx 1.5x$ attention kernel speedup at 32k-token context (Figure 4 / Table 1). We compare against flash-attn ver. 2.7.1 backend. $S^2$ANTA-flash shows $1.25\times$ decode latency speedup, and $S^2$ANTA-prop shows $1.229\times$ speedup.
>
> **Table R6: Llama 8B: Prefill, decode, and time per output token speedups versus FlashAttention-2.**
> |Method|Avg prefill/batch(s)|Avg decode/batch(s)|Time/output token(ms)|Speedup vs. FA2|
> |-|-:|-:|-:|-:|
> |FlashAttn-2|56.69|101.22|8.237|1.000×|
> |S$^2$ANTA-flash ($S=2048$)|56.61|80.97|6.589|1.250×|
> |S$^2$ANTA-prop ($S=128$)|57.03|82.36|6.702|1.229×|
>
> Results are averaged over 9 batches. The observed $1.25\times$ speedup reflects Amdahl's law for memory-bound autoregressive decoding. Assuming latency is dictated by memory bandwidth: in bf16 precision, loading Llama 8B model weights accounts for $\approx15$ GB, while a batch of six 30k-token prompts occupies $\approx24$ GB of KV cache. Our standalone kernel speedup in this regime is $\approx 1.5\times$ (our headline result in Fig. 4).
>
> If $p$ is the proportion of bandwidth occupied by KV cache and $s$ is the speedup in loading KV cache by virtue of SANTA, Amdahl's law predicts: Speedup $= 1 / ((1 - p) + p / s) = 1 / ((15/39) + (24/39) / 1.5) \approx 1.258$, which is close to our measurement.
>
> Larger batches may increase end-to-end speedup, while smaller batches may decrease gains. We demonstrate what is achievable on our 48GB GPUs.
>
> ### Response to Weakness 2: Decode-Phase Specialization
>
> The reviewer is correct: SANTA is most useful in long-context, decode-heavy workloads. We therefore position it as a specialized backend for that regime, analogous to how current systems already switch kernels across prefill/decoding and short/long-context settings.
>
> ### Response to Weakness 3/Question 5: On Datacenter GPUs
>
> Indeed, an H100 has a higher bandwidth-to-compute ratio than an RTX 6000. However, since we are focused on the memory-bound decoding regime where memory bandwidth the bottleneck anyway (and not compute), we expect the qualitative benefit to persist on datacenter GPUs, though absolute gains may vary with bandwidth, cache hierarchy, and batching. Datacenter GPUs may allow more aggressive batching, which may expand the proportion of the decode computation that SANTA targets in Amdahl's law.
>
> ### Response to Question 1: "S2ANTA-strat" and "S2ANTA-sys" naming convention
>
> $S^2$ANTA-strat and $S^2$ANTA-sys are mathematical/conceptual algorithms, whereas $S^2$ANTA-flash and $S^2$ANTA-prop are the $S^2$ANTA-sys method re-engineered as GPU kernels to exploit GPU parallelism. Figure 4 and Table 1 use these GPU kernels. We apologize if naming conventions have caused any confusion. We will clarify this in revision.
>
> ### Response to Question 2: Benchmark dtype
>
> Our benchmarks use bf16 precision for KV cache and model weights.
>
> ### Response to Question 3: On KV Cache Quantization and Amdahl's Law Tradeoffs
>
> The reviewer raises an interesting and valid point. Quantizing KV cache may shrink the total proportion of bandwidth occupied by KV cache, diminishing SANTA's gains per Amdahl's law. However, the same could be said of model weights: quantizing model weights may increase the share of bandwidth that SANTA saves. Aggressive model weight quantization, with recent ternary models such as BitNet being an extreme case, may swing Amdahl's law in the opposite direction, such that SANTA's sparsification optimizes a larger share of the total bandwidth. Investigating these combinations an exciting direction for future work.
>
> We briefly remark that Reviewer Dd3p brought ShadowKV to our attention, which finds that low-rank compression works well for keys but not for values. As the reviewer suggests, there may be interesting directions to explore regarding the tradeoffs of reducing bandwidth instead of quantizing.
>
> ### Response to Question 4: On End-to-End Latency and GQA group sizes
>
> Please see Response to Weakness 1.
>
> The reviewer correctly notes that large GQA groups may diminish the effective benefit of sparse V access. Baseline accuracy is in many cases recovered with very few samples relative to the sequence length - our $S^2$ANTA-prop headline numbers are $S = 128$ for 32k-token contexts - so even if GQA heads access different rows of $V$, in many cases, the union of accesses to rows of $V$ may still be sparse and yield decode latency benefit.
>
> New end-to-end results and Amdahl's law analysis will be added to the revision. We hope we have addressed the reviewer's concerns.
>
> Best regards,
>
> Authors

---

> > ### Author Rebuttal · Reviewer_Tb8z · 2026-04-02
> >
> > Thanks for the detailed responses. My concerns have been addressed. I will raise my score.

---

### Official Review · Reviewer_aHLS · 2026-03-11

**Soundness:** 2
**Presentation:** 2
**Significance:** 2
**Originality:** 2
**Overall Recommendation:** 4
**Confidence:** 3

**Summary:**

This paper proposes SANTA, a stochastic approximation to the value aggregation stage of attention for long-context autoregressive decoding. Instead of reading all value vectors from the KV cache, it samples a small number of value rows from the post-softmax attention distribution and averages them, yielding an unbiased estimator. The paper also introduces S²ANTA for variance reduction via stratified/systematic sampling, and a complementary Bernoulli method for sparsifying score computation. Experiments show up to 1.5× kernel-level speedup over optimized decoding baselines while maintaining similar accuracy on several benchmarks.

**Compliance With Llm Reviewing Policy:**

Affirmed.

**Final Justification:**

The rebuttal addresses my main concerns by adding both long-context evaluation and end-to-end decoding results in a realistic serving setup. These additions substantially strengthen the empirical case for the paper, so I raise my Overall Recommendation to weak accept.

**Key Questions For Authors:**

1. The method is motivated by memory-bound long-context decoding, yet most of the accuracy evaluation is conducted on relatively short-context benchmarks such as GSM8K and MMLU. Could the authors provide additional experiments demonstrating that model quality is preserved on long-context reasoning or generation tasks where the proposed method is expected to deliver the largest efficiency benefits?

2. The paper reports kernel-level speedups, but it is unclear how much improvement this yields in a full inference pipeline. Could the authors provide end-to-end decoding benchmarks (e.g., tokens/sec or latency) when integrating the method into a realistic LLM serving setup?

**Limitations:**

yes

**Strengths And Weaknesses:**

Strengths
1. Focusing on an important and practical bottleneck: memory-bound long-context decoding.
2. Combines theory with CUDA kernel design and empirical results.
3. Appears a different view from methods like KV compression and quantization.

Weaknesses
1. This paper argues that the method is most useful in memory-bound long-context decoding, yet a substantial part of the quality evaluation is on short-context benchmarks such as GSM8K and MMLU, where the motivating bottleneck is much less relevant. As a result, the experiments do not fully establish that the method preserves model quality precisely in the regime where its efficiency gains matter most.
2. The method still requires computing the full attention score distribution before sampling, so it only reduces the value-stage cost and does not fully address the overall attention bottleneck.
3. The reported gains are primarily kernel-level rather than end-to-end system speedups, making it difficult to assess the practical benefit in real inference deployments.
4. The approach introduces a sampling budget hyperparameter, and performance appears sensitive to this choice, which may require nontrivial tuning across tasks and sequence lengths.

---

> ### Author Rebuttal · Authors · 2026-03-31
>
> We thank the reviewer for their thoughtful feedback and engagement.
>
> ### Response to Weakness 1: On Long-Context Model Quality
>
> We agree with the reviewer that additional long-context benchmarks will better validate SANTA. At the reviewer's suggestion, we provide additional benchmarking on LongBench v2:
>
> **Table R4: LongBench v2**
>
> Accuracy and average tokens on LongBench v2 [1] across different sampling budgets ($S$). Accuracy shows 95% bootstrap confidence intervals.
> |Method|S|Accuracy(%)|Avg scored toks|
> |-|-:|-:|-:|
> |S²ANTA-flash|128|8.748±2.964|30035.5|
> |S²ANTA-flash|256|14.049±3.504|30024.0|
> |S²ANTA-flash|512|26.574±3.740|29990.0|
> |S²ANTA-flash|1024|27.170±1.509|29994.2|
> |S²ANTA-flash|2048|28.827±3.422|29996.4|
> |S²ANTA-prop|64|29.092±2.534|29997.5|
> |S²ANTA-prop|128|28.098±1.028|29998.4|
> |S²ANTA-prop|256|28.694±1.029|29995.4|
> |S²ANTA-prop|1024|28.562±1.588|29996.2|
> |SDPA|—|28.363±1.509|29996.0|
>
> Across 3 runs, prompts longer than 32k are truncated to 32k following the repository protocol. In this 30k-token regime, $S^2$ANTA-flash ($S=2048$) and $S^2$ANTA-prop ($S=128$) recover SDPA accuracy, matching the regime where Fig. 4 reports $\approx1.5\times$ kernel speedup.
>
> ### Response to Weakness 2: On Computing the Attention Score
>
> The reviewer is correct that SANTA leaves the score stage $qK^\mathsf{T}$ unchanged. Our claim is narrower: SANTA is a modular optimization of softmax-$V$, complementary to score-stage methods (e.g., low-rank compression or quantization) and to cache-retention/eviction policies. We frame SANTA not as an end-to-end replacement for efficient attention methods, but as a modular optimization that could be implemented in tandem with other methods.
>
> ### Response to Weakness 3: On End-to-End Speedups
>
> We integrated our kernels with HuggingFace's generate() function, showing end-to-end speedups for batch inference. We consider representative QA prompts, truncated to 30k tokens for batch inference, for a batch size of 6 on RTX 6000 Ada, decoding 2k new tokens. We set sample budgets $S$ matching our headline results which show $\approx 1.5x$ attention kernel speedup at 32k-token context (Figure 4 / Table 1). We compare against a flash-attn ver. 2.7.1 backend. $S^2$ANTA-flash shows $1.25\times$ decode latency speedup, and $S^2$ANTA-prop shows a $1.229\times$ speedup.
>
> **Table R6: Llama 8B: Prefill, decode, and time per output token speedups versus FlashAttention-2.**
> |Method|Avg prefill/batch(s)|Avg decode/batch(s)|Time/output token(ms)|Speedup vs. FA2|
> |-|-:|-:|-:|-:|
> |FlashAttn-2|56.69|101.22|8.237|1.000×|
> |S$^2$ANTA-flash ($S=2048$)|56.61|80.97|6.589|1.250×|
> |S$^2$ANTA-prop ($S=128$)|57.03|82.36|6.702|1.229×|
>
> Results are averaged over 9 batches. The observed $1.25\times$ speedup closely reflects Amdahl's law for memory-bound autoregressive decoding. We assume that latency is dictated by memory bandwidth and bytes moved. In bf16 precision, loading Llama 8B model weights accounts for $\approx15$ GB, while a batch of six 30k-token prompts occupies $\approx24$ GB of KV cache. Our standalone kernel speedup in this regime is $\approx 1.5\times$ (which is our headline result in Figure 4).
>
> If $p$ is the proportion of bandwidth occupied by KV cache and $s$ is the speedup in loading KV cache by virtue of SANTA, then Amdahl's law would predict: Total Speedup $= 1 / ((1 - p) + p / s) = 1 / ((15/39) + (24/39) / 1.5) \approx 1.258$, which is close to our observed end-to-end speedup.
>
> We briefly remark that larger batches may increase end-to-end speedup, while smaller batches may decrease gains. We demonstrate the achievable end-to-end speedup on the 48GB GPUs used for this study. Since SANTA is modular and operates only on the softmax-$V$ accumulation, it could in principle be combined with KV cache eviction policies, CPU offloading techniques, or quantization for potential multiplicative efficiency gains.
>
> ### Response to Weakness 4: On sample budget parameter tuning
>
> The reviewer correctly notes that SANTA has a sample budget hyperparameter $S$. While the scope of this paper is establishing SANTA's viability, we believe tuning the sample budget is an exciting direction for future work. We briefly remark that Table R1 (in our response to Reviewer Dd3p) suggests that peaky/low entropy attention distributions may accommodate fewer samples; dynamically allocating samples at runtime may unlock further efficiency gains than our static allocation. In this work, we show that even with minimal tuning, we can still achieve $1.5\times$ kernel speedup ($1.25\times$ end-to-end) at 32k-token contexts while retaining baseline accuracy on LongBench, NIAH, FWE, and QA tasks.
>
> Q1 and Q2 are addressed by W1 and W3, respectively.
>
> New LongBench results, end-to-end decode latency, and Amdahl's law analysis will be added to the revision. We hope these new empirical results address the core of your critique.
>
> Best regards,
>
> Authors

---

> > ### Author Rebuttal · Reviewer_aHLS · 2026-04-05
> >
> > My concerns have been addressed. I will raise my score.

---

### Official Review · Reviewer_2Gn9 · 2026-03-12

**Soundness:** 3
**Presentation:** 2
**Significance:** 2
**Originality:** 3
**Overall Recommendation:** 3
**Confidence:** 4

**Summary:**

In this paper, the authors investigate the problem of memory-bandwidth bottlenecks in long-context autoregressive decoding for transformer-based large language models, where each decoding step requires streaming the full key–value (KV) cache. To alleviate this limitation, the authors propose Stochastic Additive No-mulT Attention (SANTA), a stochastic sparse attention mechanism that approximates the post-softmax value aggregation by sampling a small number of value rows from the attention distribution instead of accessing the entire value cache. To improve estimation quality, the authors further introduce S²ANTA, a variance-reduced variant based on stratified and systematic sampling strategies. In addition, the paper proposes Bernoulli (qK^T) sampling, which represents query elements as stochastic ternary variables to induce sparse key-feature access during score computation. The proposed methods reduce KV memory access and replace dense multiply–accumulate operations with gather-and-add computations while maintaining unbiased attention estimation. Efficient GPU kernels are designed to parallelize the sampling and sparse aggregation processes. Extensive experiments on long-context benchmarks and reasoning tasks demonstrate that the proposed approach achieves notable decoding speedups while preserving model accuracy.

**Compliance With Llm Reviewing Policy:**

Affirmed.

**Final Justification:**

The paper proposes a stochastic sparse attention mechanism for accelerating long-context decoding. However, I remain unconvinced about its practical relevance, and my concerns were not satisfactorily addressed in the rebuttal.

**Lack of evaluation with KV selection methods.** The paper claims SANTA is complementary to KV selection approaches, but does not provide experiments combining them. Moreover, no additional experiments involving KV selection baselines were provided during the rebuttal. Since KV selection directly reduces effective context size, it is unclear whether SANTA provides meaningful additional benefits in realistic inference pipelines.

**Unclear behavior across context lengths.** SANTA underperforms top-k attention in short-context settings (Table 11), yet neither the paper nor the rebuttal explains this behavior. While improvements are shown on specific long-context tasks, it remains unclear under what conditions SANTA consistently outperforms top-k or similar approaches.

Overall, the current empirical evidence is insufficient to justify the claimed effectiveness and practical value of the method, and I maintain my original recommendation.

**Key Questions For Authors:**

1.	Provide quantitative analysis of the numerical approximation error introduced by stochastic sampling (e.g., $|AV - \hat{A}V|$ or cosine similarity).
2.	Include comparisons with additional sparse attention or dynamic context selection methods.
3.	Investigate hybrid strategies that deterministically preserve high-attention tokens while sampling the remaining ones.
4.	Evaluate the method on a broader set of reasoning and coding benchmarks to better demonstrate general applicability.

**Limitations:**

Yes.

**Strengths And Weaknesses:**

The paper studies stochastic sparse attention for improving the efficiency of long-context LLM decoding and proposes a sampling-based approximation of the attention aggregation.

Positive aspects of the work include the following.
1.	The paper proposes a stochastic sampling-based attention approximation that replaces dense value aggregation with sampled value accumulation. Interpreting attention weights as a probability distribution and estimating the attention output through sampling is conceptually simple and well motivated, and the proposed S²ANTA variant further reduces variance through stratified sampling.
2.	The work addresses the practically important problem of memory-bandwidth bottlenecks in long-context LLM inference. The authors implement custom CUDA kernels and demonstrate measurable speed improvements in attention kernels under long-context settings, suggesting potential practical value for improving decoding efficiency.

The work also has several limitations that could be further addressed.
1.	The paper mainly evaluates the method through downstream task accuracy, but does not provide a quantitative analysis of the numerical approximation error introduced by stochastic sampling. Providing metrics such as $|AV - \hat{A}V|$, cosine similarity, or relative error between dense and approximated attention outputs would help better characterize the fidelity of the approximation and the relationship between sampling budget and estimation quality.
2.	The experimental comparisons with existing sparse attention methods are relatively limited. Although top-$k$ attention is discussed and partially evaluated in the appendix, the main experiments do not include systematic comparisons with commonly used sparse attention baselines under comparable sparsity ratios or computational budgets. In particular, comparisons with recent dynamic context selection or sparse inference approaches such as OmniKV [1] and DuoAttention [2] would help better position the contribution.
3.	Although the estimator is theoretically unbiased, it is unclear whether unbiased sampling provides the most suitable approximation for attention computation in practice. Since tokens with larger attention scores contribute more to the output in $AV = \sum_i A_i V_i$, missing such tokens during sampling may introduce larger errors than missing low-weight ones. Additional empirical analysis would help support the claim that unbiased stochastic sampling is preferable.
4.	The stochastic sampling mechanism introduces variance in the attention estimation, and tokens with large attention weights may occasionally not be selected. The paper does not discuss whether hybrid strategies—such as deterministically preserving tokens with attention weights above a threshold while sampling the remaining ones—could reduce variance while retaining computational efficiency.
5.	The experimental evaluation is conducted on a relatively limited set of benchmarks, mainly including GSM8K, MMLU, and several long-context QA tasks. Since the proposed method is intended as a general decoding strategy for long-context LLM inference, additional evaluation on a broader range of reasoning-intensive and domain-diverse benchmarks would help better demonstrate its general applicability. For example, datasets such as OlympiadBench [3], MATH [4], LiveCodeBench [5], or HumanEval [6] could provide a more comprehensive assessment across reasoning and coding scenarios.

References
[1] OmniKV: Dynamic Context Selection for Efficient Long-Context LLMs.
[2] DuoAttention: Efficient Long-Context LLM Inference with Retrieval and Streaming Heads.
[3] OlympiadBench: A Challenging Benchmark for Promoting AGI with Olympiad-Level Bilingual Multimodal Scientific Problems. arXiv, 2024.
[4] Solving Quantitative Reasoning Problems with Language Models. NeurIPS, 2022.
[5] LiveCodeBench: Holistic and Contamination-Free Evaluation of Large Language Models for Code. ICLR, 2025.
[6] Evaluating Large Language Models Trained on Code. arXiv, 2021.

---

> ### Author Rebuttal · Authors · 2026-03-31
>
> We thank the reviewer for the thoughtful and technically sharp comments.
>
> ### Response to Weakness 1: On Numerical Approximation Error
>
> **Table R3: $S^2$ANTA-flash & $S^2$ANTA-prop numerical approximation error**
> |Task|Backend|S|Mean Relative L2 Error|Mean Cosine Similarity|Same-State Top-1 Next-Token Agreement|
> |-|-|-|-|-|-|
> |FWE|flash|256|.244|.963|91%|
> |FWE|flash|1024|.091|.993|97%|
> |FWE|flash|2048|.055|.997|97%|
> |FWE|prop|128|.131|.986|96%|
> |FWE|prop|256|.085|.994|97%|
> |FWE|prop|1024|.035|.999|98%|
> |QA2|flash|256|.572|.851|91%|
> |QA2|flash|1024|.233|.968|97%|
> |QA2|flash|2048|.144|.986|99%|
> |QA2|prop|128|.208|.967|99%|
> |QA2|prop|256|.141|.983|97%|
> |QA2|prop|1024|.063|.996|99%|
>
> We added a same-state fidelity analysis on 32k-token FWE and QA2 prompts. Relative L2 error and cosine similarity are computed only on the final decoding layer's attention output $AV$, comparing sparse and dense attention at matched decode states. By contrast, top-1 agreement is computed from a full one-token model forward pass and measures whether the approximate model preserves the dense model's top-1 next-token prediction. Approximation quality improves with budget, and the low 91\% top-1 agreement at flash $S=256$ is consistent with its downstream accuracy drop (Table 1), while the paper's operating points ($S=2048$ flash, $S=128$ prop) recover high fidelity and high next-token agreement.
>
> ### Response to Weakness 2: On Sparse Attention Baselines
>
> We appreciate the reviewer highlighting DuoAttention and OmniKV and will add them as related references. DuoAttention partitions heads into retrieval vs. streaming heads (full KV vs. sinks/recent tokens), while OmniKV retains/offloads the full KV, using filter layers to select a decode-time subset for later layers. SANTA, instead, fundamentally modifies the softmax-$V$ operator itself.
>
> As the reviewer insightfully suggests, deterministic selection and stochastic aggregation are not mutually exclusive: SANTA could, in principle, be applied to DuoAttention’s retrieval heads or to OmniKV’s selected subset, replacing dense softmax-$V$ aggregation in either case. We therefore position SANTA as a modular, complementary tool to system-level CPU–GPU offloading and KV-cache retention methods, rather than a direct, like-for-like comparison for the scope of this paper.
>
> ### Response to Weaknesses 3 and 4: On Variance Reduction and Hybrid Deterministic + Sampling Strategies
>
> The reviewer raises a subtle and important point: unbiasedness alone is not sufficient; low variance is critical in practice. This is why we move from i.i.d. SANTA to the variance-reduced S²ANTA in our paper: stratified sampling has a formal variance-reduction guarantee relative to i.i.d. sampling, and systematic sampling empirically behaves similarly (Appendix D).
>
> Intuitively, by sampling once from each equal-mass CDF stratum, S²ANTA drastically reduces the probability that large-mass regions of the attention distribution are missed. For example, given attention weights [0.5, 0.3, 0.2] and a sample budget of $S=2$, one sample must fall in each half of the CDF, mathematically guaranteeing the 0.5-mass token is represented. While hybrid deterministic+stochastic policies are promising future directions, we view them as a natural extension on top of the core variance-reduced estimator established here.
>
> ### Response to Weakness 5: On Benchmark Diversity
>
> **Table R4: LongBench v2**
>
> Accuracy is reported with 95% bootstrap confidence intervals.
> |Method|S|Accuracy(%)|Avg scored toks|
> |-|-:|-:|-:|
> |S²ANTA-flash|128|8.748±2.964|30035.5|
> |S²ANTA-flash|256|14.049±3.504|30024.0|
> |S²ANTA-flash|512|26.574±3.740|29990.0|
> |S²ANTA-flash|1024|27.170±1.509|29994.2|
> |S²ANTA-flash|2048|28.827±3.422|29996.4|
> |S²ANTA-prop|64|29.092±2.534|29997.5|
> |S²ANTA-prop|128|28.098±1.028|29998.4|
> |S²ANTA-prop|256|28.694±1.029|29995.4|
> |S²ANTA-prop|1024|28.562±1.588|29996.2|
> |SDPA|—|28.363±1.509|29996.0|
>
> **Table R5: HumanEval**
> Pass@1 (%) accuracy is reported with 95% bootstrap confidence intervals.
> |Method|S|Pass@1(%)|Avg tok.|
> |-|-:|-:|-:|
> |S²ANTA-flash|16|12.805±4.370|521.6|
> |S²ANTA-flash|32|44.106±6.707|296.5|
> |S²ANTA-flash|64|59.350±6.504|267.7|
> |S²ANTA-flash|128|60.569±6.809|263.0|
> |S²ANTA-flash|256|63.008±6.911|260.1|
> |S²ANTA-prop|16|46.545±6.301|274.6|
> |S²ANTA-prop|32|61.789±6.504|267.1|
> |S²ANTA-prop|64|60.366±7.012|262.5|
> |S²ANTA-prop|128|62.398±6.911|262.5|
> |S²ANTA-prop|256|63.211±7.012|261.1|
> |SDPA|—|65.244±7.317|261.4|
>
> On LongBench v2 (30k tokens), operating points recover baseline accuracy: SDPA 28.36, flash 28.83 at $S=2048$, prop 28.10 at $S=128$. On HumanEval, we do not claim kernel speedups in the short sequence setting, but accuracy remains close to baseline: SDPA 65.24, flash 63.01, prop 63.21 at $S=256$.
>
> All results will be added to the revision. We hope this helps address the reviewer's concerns.
>
> [1] LongBench v2:Towards Deeper Understanding and Reasoning on Realistic Long-context Multitasks

---

> > ### Author Rebuttal · Reviewer_2Gn9 · 2026-04-03
> >
> > Thank you for the detailed responses. I appreciate the additional experiments and clarifications provided.
> >
> > However, the following concerns remain:
> >
> > 1. The rebuttal emphasizes variance reduction, but this does not directly address the relationship between variance and downstream performance. While S²ANTA improves over i.i.d. sampling, this does not explain why stochastic sampling should outperform deterministic selection. The empirical results (e.g., Table 11 on GSM8K) still show worse performance than top-$k$, leaving the practical advantage unclear.
> >
> > 2. The rebuttal argues that the method is complementary to KV selection approaches such as OmniKV and DuoAttention, but this positioning is not fully convincing. These methods already retain high-attention tokens and truncate low-probability regions, which may reduce the benefit of stochastic sampling. Without empirical validation, it remains unclear whether the methods are truly complementary or whether they compete in practice.
> >
> > Overall, while the rebuttal addresses several points, these core concerns remain unresolved. Therefore, I maintain my negative rating.

---

> > > ### Author Response · Authors · 2026-04-03
> > >
> > > **On stochastic sampling vs. deterministic top-k:**
> > >
> > > We note that the reviewer cites Table 11 (GSM8K, DeepSeek 7B), a short-context benchmark where we do not claim kernel speedups. In the long-context regime where SANTA delivers its efficiency gains, S²ANTA actually outperforms top-k: Table 15 (Appendix H) shows S²ANTA leading on virtually every long-context task at matched budgets (k = S), with the gap most pronounced on multi-hop QA (~3-5%). We attribute this to multi-hop reasoning requiring attention to information dispersed across the sequence, not concentrated in the top-k entries. Top-k permanently drops these tokens; stratified sampling retains nonzero probability of selecting any token.
> > >
> > > Additionally, at equal budgets, top-k costs n_q k d_k multiplications while SANTA costs zero multiplications (Table 10), yielding ~5x energy reduction in value-stage FLOPs (Section 3.5). In summary, SANTA provides an unbiased estimator that empirically outperforms top-k where it matters most (long context), while using fundamentally cheaper arithmetic.
> > >
> > > **On complementarity with KV selection methods:**
> > >
> > > We appreciate the nuanced point that KV selection methods could reduce SANTA's marginal benefit by pre-filtering to high-attention tokens. However, even after selection, the retained subset remains a dense, bandwidth-bound aggregation. In long-context settings, retained subsets can still span thousands of tokens. SANTA sparsifies the aggregation step itself, regardless of how the upstream subset was chosen.
> > >
> > > Best regards,
> > >
> > > Authors

---

### Official Review · Reviewer_Dd3p · 2026-03-13

**Soundness:** 3
**Presentation:** 3
**Significance:** 3
**Originality:** 3
**Overall Recommendation:** 5
**Confidence:** 3

**Summary:**

This work proposes a method that speeds up LLM decoding by reducing value cache access by sampling a subset of tokens from the attention distribution. They also present ways to reduce the varince of the estimation and system-level optimized kernels for tiled operations.

**Compliance With Llm Reviewing Policy:**

Affirmed.

**Final Justification:**

The rebuttal reinforced my prior assessment that the submission is a technically solid paper.

**Key Questions For Authors:**

**[Q1]** How sensitive is S^2ANTA-flash to how uniform the attention distribution is, or to the number of tiles? It would be helpful to understand how robust it is under those chnages.

**Limitations:**

yes

**Strengths And Weaknesses:**

### Strengths

* Clear motivation and well-justified design choices.
* Practically valuable because it is orthogonal to KV cache compression appraoches such as hard dropping and quantization.
* Good systems-level consideration, especially the tiling-aware design in the spirit of FlashDecoding.
* Strong analysis of where the speedup comes from.

### Weaknesses

**[W1]** The empirical validation could be strengthened with more relevant selective-retrival basleines. In particular, ShadowKV seems like an important missing baseline.

---

> ### Author Rebuttal · Authors · 2026-03-31
>
> We sincerely thank the reviewer for the thoughtful feedback.
>
> ### Response to Weakness 1
>
> > The empirical validation could be strengthened with more relevant selective-retrival basleines. In particular, ShadowKV seems like an important missing baseline.
>
> Thank you for pointing out ShadowKV. We agree it is an important adjacent method and will certainly cite and discuss it in the revision. ShadowKV is a system-level, long-context inference method that stores low-rank pre-RoPE keys plus landmarks/outliers on the GPU, offloads the value cache to the CPU, and performs decode-time chunk selection, value fetching, and key reconstruction.
>
> By contrast, our main decode-speedup results target the softmax–V (value-aggregation) operator itself. We therefore view ShadowKV as complementary rather than a direct, apples-to-apples baseline for the core operator-level kernel studied here. More broadly, we frame SANTA as a modular operator-level primitive that can compose with system-level KV retention, offloading, or selection methods, rather than as a replacement for them. A natural hybrid would be to apply SANTA to replace dense softmax-V aggregation over ShadowKV’s selected/reconstructed sparse KV subset. At the operator level, top-k attention is the closer deterministic comparator, which we briefly analyze in Appendix H.
>
> ### Response to Question 1
>
> > How sensitive is S$^2$ANTA-flash to how uniform the attention distribution is, or to the number of tiles? It would be helpful to understand how robust it is under those chnages.
>
> At the reviewer's suggestion, we performed additional analyses to characterize how the approximation quality of S$^2$ANTA-flash varies with (i) the spread of attention mass across tiles and (ii) the selected tile size. For both frequent-word extraction (FWE) and multi-hop QA (QA2), we selected 40 long prompts (32k tokens) and examined 32 decoding steps per prompt. At each step, using the dense model's current prefix as the reference context, we applied S$^2$ANTA-flash only in the final decoding layer and compared the resulting attention output at that layer with the corresponding dense-attention output. Reported relative L2 error and cosine similarity are measured at the final decoding layer and averaged over attention heads and the 32 evaluated decoding steps.
>
> We fix the S$^2$ANTA-flash sample budget to $S=2048$, which matches the main operating point in the paper where S$^2$ANTA-flash recovers baseline accuracy at 32k-token contexts while delivering the reported $1.5\times$ kernel-level speedup.
>
> **Table R1: Sensitivity of S$^2$ANTA-flash to attention support spread across tiles.**
> | Task | Tiles to Reach 90% Attention Score Mass| Mean Relative L2 Error | Mean Cosine Similarity |
> | --- | --- | --- | --- |
> | FWE | 1 tile | 0.035 | 0.998 |
> | FWE | 2 tiles | 0.047 | 0.998 |
> | FWE | 3-4 tiles | 0.068 | 0.997 |
> | FWE | 5-8 tiles | 0.084 | 0.995 |
> | FWE | 9+ tiles | 0.073 | 0.996 |
> | QA2 | 1 tile | 0.055 | 0.997 |
> | QA2 | 2 tiles | 0.117 | 0.991 |
> | QA2 | 3-4 tiles | 0.170 | 0.982 |
> | QA2 | 5-8 tiles | 0.175 | 0.981 |
> | QA2 | 9+ tiles | 0.190 | 0.978 |
>
> For Table R1, we set the tile size to 256 (the same setting used in the paper). At each exact decode position, we compute the dense attention distribution and count how many tiles are needed to capture 90% of attention weights. We then bucket observations by this quantity and report the corresponding L2 error and Cosine similarity. S$^2$ANTA-flash is most accurate when most attention mass is concentrated in a small number of tiles, and degrades as the distribution becomes more diffuse. In our study, we established that even with minimal tuning, setting $S = 2048$ recovers baseline accuracy at 32k-token contexts. Future work may tune sample allocation according to attention distribution statistics for further gains.
>
> **Table R2: Sensitivity of S$^2$ANTA-flash to tile size.**
>
> | Task | Tile Size | Mean Relative L2 Error | Mean Cosine Similarity |
> | --- | --- | --- | --- |
> | FWE | 32 (many tiles) | 0.145 | 0.985 |
> | FWE | 64 | 0.119 | 0.989 |
> | FWE | 256 (fewer tiles) | 0.055 | 0.997 |
> | QA2 | 32 (many tiles) | 0.390 | 0.918 |
> | QA2 | 64 | 0.269 | 0.958 |
> | QA2 | 256 (fewer tiles) | 0.144 | 0.986 |
>
> We sweep $S^2$ANTA-flash tile size using the same protocol. Given a fixed 32k sequence length, smaller tile size implies a greater number of tiles in total. Table R2 shows that larger tiles are substantially more faithful than smaller ones. Our intuition is that small tile sizes are likely to assign sample budget to tiles with small or negligible attention weights, thereby "wasting" samples on parts of the attention distribution with low attention scores.
>
> ---
> We will add tile size and attention distribution uniformity experiments to the revision. We hope these analyses address the reviewer's concerns.
>
> Best regards,
>
> Authors

---

> > ### Author Rebuttal · Reviewer_Dd3p · 2026-04-03
> >
> > Thank you for the effort in the response.
> > Please include the additional experimental results in the revision.
> > I will raise my score.

---

### Decision · Program_Chairs · 2026-04-30

**Decision:**

Accept (regular)

**Comment:**

This paper proposes an attention acceleration method. The main idea is to speed up attention weight * value multiplication with sampling. Reviewers acknowledge the paper quality. However, there also consideration remains: including relationship/advantage/combinability of the proposed method with existing sparse attention/kv selection methods, whether stochastic selection is in principle better/worse than deterministic selection, the inferior performance than topk on gsm8k, and the lack of speedup results on mainstream datacenter gpus.

As the positive review outweighs negative opinion, I still recommend acceptance. However, reviewer's concerns are also valid. I understand that the concerns are not easy to fixed during rebuttal due to the short period and limited reply length. Please carefully read reviewer's comments and improve the paper based on the comments in the final version.

Though not raised by any reviewer, AC personally consider multi-latent attention (MLA) as a baseline as well, as the proposed approach treats K and V differently, it may not improve MLA, which is bandwidth efficient anyway. This personal consideration does not affect AC's recommendation.